# Protein phosphatase 2A inactivation induces microsatellite instability, neoantigen production and immune response

Yu-Ting Yen[1,2], May Chien[1,2], Pei-Yi Wu[1,2], Chi-Chang Ho[1,2], Chun-Te Ho[1,2], Kevin Chih-Yang Huang [3,4], Shu-Fen Chiang[5,6], K. S. Clifford Chao[5], William Tzu-Liang Chen[7,8] & Shih-Chieh Hung [1,2,9 ✉]

Microsatellite-instable (MSI), a predictive biomarker for immune checkpoint blockade (ICB) response, is caused by mismatch repair deficiency (MMRd) that occurs through genetic or epigenetic silencing of MMR genes. Here, we report a mechanism of MMRd and demonstrate that protein phosphatase 2A (PP2A) deletion or inactivation converts cold microsatellite-stable (MSS) into MSI tumours through two orthogonal pathways: (i) by increasing retinoblastoma protein phosphorylation that leads to E2F and DNMT3A/3B expression with subsequent DNA methylation, and (ii) by increasing histone deacetylase (HDAC)2 phosphorylation that subsequently decreases H3K9ac levels and histone acetylation, which induces epigenetic silencing of MLH1. In mouse models of MSS and MSI colorectal cancers, triple-negative breast cancer and pancreatic cancer, PP2A inhibition triggers neoantigen production, cytotoxic T cell infiltration and ICB sensitization. Human cancer cell lines and tissue array effectively confirm these signaling pathways. These data indicate the dual involvement of PP2A inactivation in silencing MLH1 and inducing MSI.

[1] Drug Development Center, Institute of New Drug Development, China Medical University, Taichung 40402, Taiwan. [2] Integrative Stem Cell Center, China Medical University Hospital, Taichung 40402, Taiwan. [3] Department of Biomedical Imaging and Radiological Science, China Medical University, Taichung 40402, Taiwan. [4] Translation Research Core, China Medical University Hospital, China Medical University, Taichung 40402, Taiwan. [5] Proton Therapy and Science Center, China Medical University Hospital, China Medical University, Taichung, Taiwan 40402, ROC. [6] Lab of Precision Medicine, Feng-Yuan Hospital, Ministry of Health and Welfare, Taichung 42055, Taiwan. [7] Department of Colorectal Surgery, China Medical University Hospital, China Medical University, Taichung 40402, Taiwan. [8] Department of Colorectal Surgery, China Medical University HsinChu Hospital, China Medical University, HsinChu 302, Taiwan. [9] Department of Orthopaedics, China Medical University Hospital, Taichung 40402, Taiwan. ✉email: hung3340@gmail.com

Microsatellite-instable (MSI) tumours with defective DNA mismatch repair (MMR)[1] account for 15% of sporadic colorectal cancers (CRCs)[2], which mainly occur through epigenetic silencing that inactivates the somatic biallelic of the MMR genes[3]. MSI is associated with better stage-adjusted prognosis[4] in early stage I–III CRC[4] and response to immune checkpoint blockade (ICB)[5] than microsatellite-stable (MSS) tumours, leading to the urgent need to investigate the mechanisms causing MSI tumour development.

The programmed death 1 (PD-1) pathway, a negative feedback system that represses Th1 cytotoxic immune responses, is upregulated in many tumours. ICB with antibodies against PD-1 or its ligand (PD-L1) has led to remarkable clinical responses in patients with different types of cancers, including melanomas, non–small-cell lung cancer, renal cell carcinoma, bladder cancer, and Hodgkin's lymphoma[6–12]. Its success or failure is mainly determined by tumour intrinsic factors. MMR defects lead to accumulation of mutation burden, a potential source of immunogenic neoantigens recognised by the immune system, with high immunoscores and cytotoxic T cell infiltration[13]. Indeed, CRC and non-CRC patients with MMR deficiency (MMRd) have better responses to PD-1 ICB therapy and show improved progression-free survival[14]. However, the molecular basis of these clinical features is poorly understood.

The presence of a BRAF (V600E) mutation promotes MLH1 silencing by increasing the levels of the v-maf avian musculoaponeurotic fibrosarcoma oncogene homologue G at the promoters of MLH1[15]. However, this cannot explain the mechanism of most MSI CRC cases that lack this mutation. Moreover, the activating mutation in the oncogene BRAF is genetically seen subsequent to MLH1 hypermethylation[16]. Emerging evidence also suggests that certain miRNAs can regulate MMR expression to influence genomic stability in CRC[17,18]; however, this process is not through epigenetic silencing of MLH1.

Protein phosphatase 2A (PP2A), consisting of a catalytic subunit (C), a structural subunit (A), and a regulatory/variable B-type subunit, is a major serine/threonine phosphatase in cells and maintains cell homoeostasis by counteracting most of the kinase-driven signalling pathways. Targeting PP2A as a therapeutic strategy has recently gathered much attention[19,20], yet the complexity of cellular functions and pathways regulating PP2A has meant that the anti-tumour effects of activating[19] or inhibiting[20] PP2A activity remain poorly elucidated. Emerging evidence indicates that PP2A inactivation, caused either by mutations or by endogenous inhibitors, has a major role in the maintenance of the transformed phenotype in cancer, but its distinct tumour-intrinsic role in shaping the immune microenvironment and response to ICB remains unknown.

In this work, we report a mechanism of MMRd and demonstrate that PP2A deletion or inactivation convert cold MSS into MSI tumours with increased cytotoxic T cell infiltration and improved response to ICB in pre-clinical cancer models.

## Results

### Loss or inhibition of PP2A enhances cytotoxic T cell infiltration, inhibits regulatory T cell infiltration, and correlates with MSI status.
To provide insights into the inherent role of PP2A inactivation in promoting immune cell infiltration in CRC, we first demonstrated that mouse intestinal tumours, developed by the conditional deletion of Ppp2r1a (gene encoding PP2A scaffold protein in 95%) in Lgr5+ crypt stem cells (referred to as Ppp2r1a$^{-/-}$)[21], increased infiltration of CD8+ T cells and CD20+ B cells but decreased FOXP3+ regulatory T cell (Treg) infiltration (Fig. 1a) and enriched cytokine, chemokine, interferon (IFN)-γ, and JAK-STAT

pathways (Fig. 1b). In human colon cancers (based on The Cancer Genome Atlas (TCGA) data), the expression of endogenous PP2A inhibitors, CIP2A and SET, correlated positively with the infiltration of CD8+ T cells and CD20+ B cells and negatively with FOXP3+ Treg cells (Fig. 1c). Contrastingly, the expression of PPP2R1A correlated negatively with CD8+ T cells but positively with FOXP3+ Treg cells (Fig. 1c). Similar findings were observed in human rectal cancers (TCGA; Fig. 1d). Besides, the expression of some of these PP2A modulators positively or negatively correlated with other immune cells, such as macrophages, neutrophils, and dendritic cells, both in human colon and rectal cancers (Supplementary Fig. 1). These data indicate that PP2A inactivation in murine and human CRC shapes the immune microenvironment, including promoting cytotoxic T or B cell infiltration and inhibiting Treg cell infiltration in tumours.

Human MSI CRC displays mRNA expression signatures characteristic of increased immunogenicity, including upregulation of pro-inflammatory cytokines and cytotoxic mediators[22]. Analysis of human CRC TCGA data further revealed increased CD8+ T cell infiltration in MSI compared to MSS (Supplementary Fig. 2), supporting the possibility that increased immunogenicity and cytotoxic T cell infiltration in PP2A-inactivated tumours were mediated by the induction of MSI status. As expected, we observed that mouse Ppp2r1a$^{-/-}$ intestine tumours displayed high mutation burden (Fig. 2a), MMR-deficient signatures (Fig. 2b), COSMIC signatures associated with defective MMR (Fig. 2c), positive MSI assay results (Fig. 2d), downregulated MMR-associated genes (Fig. 2e), and reduced MLH1 protein levels in Ppp2r1a$^{-/-}$ intestine tumour organoid cultures (Fig. 2f, g) and primary tumour tissues (Fig. 2h). Analysis of human CRC TCGA data (Fig. 2i) and immunohistochemical profiling by tissue array (Fig. 2j and Supplementary Table 7) also revealed that MSI increased both the mRNA and protein levels of CIP2A and SET but decreased the PPP2R1A level compared to those of MSS (Fig. 2i). These data together suggest that PP2A inactivation induces MSI status in murine and human CRC.

### PP2A inactivation silences MLH1 via the Rb-E2F-DNMT3A/3B and HDAC2-H3K9ac epigenetic pathways.
Since epigenetic inactivation of MLH1 and DNA methylation cause most human MSI CRC[2], we first checked the promoter DNA methylation in mouse Ppp2r1a$^{-/-}$ intestinal tumours. Both methylation-specific polymerase chain reaction (Supplementary Fig. 3a), and bisulfite sequencing (Supplementary Fig. 3b) demonstrated increased methylation of CpG islands of MLH1 in Ppp2r1a$^{-/-}$ intestinal tumours compared to controls. Moreover, RNA-seq analysis (Supplementary Fig. 3c), followed by confirmation with western blotting (Supplementary Fig. 3d) revealed that Ppp2r1a deletion-induced MLH1 epigenetic silencing was associated with upregulation of the de novo DNA methyltransferases DNMT3A and DNMT3B. Although PP2A regulates pathways involved in the cell cycle, metabolism, migration, and survival[19], little, if any, is known about its role in epigenetic regulation. To identify the binding partners of Ppp2r1a for this function, we performed immunoprecipitation followed by mass spectrometry (IP/MS) in normal intestinal organoid cultures and analysed the enriched pathways by comparing transcriptomes between normal and Ppp2r1a$^{-/-}$ intestinal tumour organoid cultures (Fig. 3a). IP/MS identified several known proteins, such as protein phosphatase 2 catalytic subunit beta (PPP2CB), retinoblastoma protein (Rb)[23] and AKT1[24] in ovarian carcinoma and HEK cells and unknown proteins, such as histone deacetylase (HDAC)2, interacted with and regulated by Ppp2r1a (Supplementary Data 1). RNA-seq and gene set enrichment analysis (GSEA) revealed several pathways upregulated or downregulated in Ppp2r1a$^{-/-}$ intestinal tumour organoid cultures,

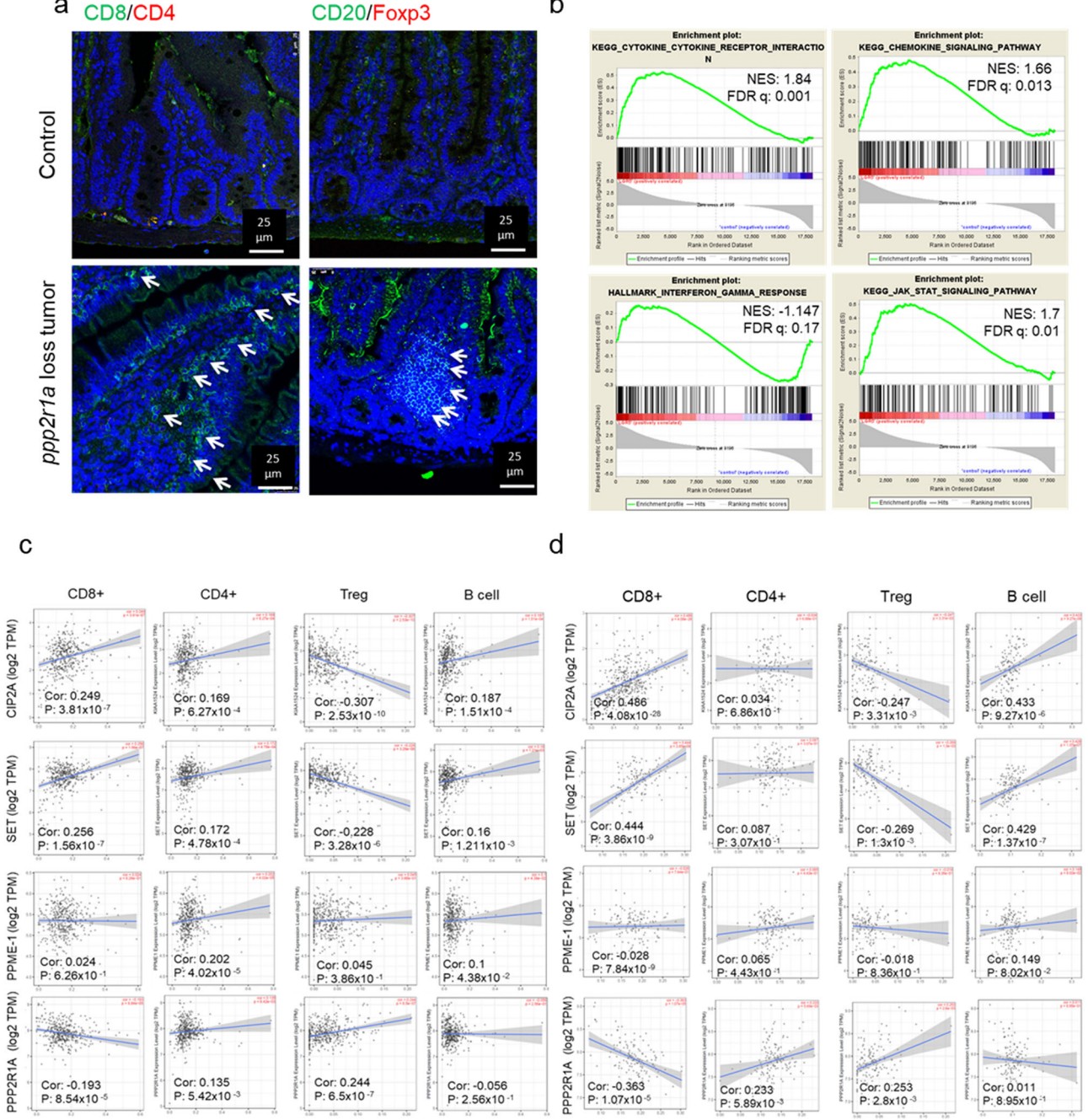

**Fig. 1 PP2A loss or inhibition enhances cytotoxic T cell infiltration in colorectal cancer. a**, **b** Ppp2r1a-loss colon tumours were induced in *Lgr5-EGFP-CreERT2; Ppp2r1a*$^{flox/flox}$ by treatment with DMBA and tamoxifen for 36 days. Control colon tissues (Control) and Ppp2r1a-loss colon tumours were harvested for analysis. **a** Representative images of immunofluorescence showing increased CD8+ and CD20+ and decreased Foxp3+ infiltration in murine Ppp2r1a-loss colon tumours compared to controls. Arrows indicate positive signals. Images are representative of three biological independent samples for each group. Bar = 25 μm. **b** Gene set enrichment analysis (GSEA) for cytokine, chemokine, IFN-gamma response, and JAK-STAT signalling pathways in murine Ppp2r1a-loss colon tumours compared to controls. **c** TCGA-COAD (*n* = 461) and **d** TCGA-READ (*n* = 172) omics analysis revealed that CIP2A and SET levels were significantly positively correlated with CD8+, CD4+, and B cell infiltration levels and significantly negatively correlated with Treg infiltration, while PPP2R1A levels were negatively correlated with CD8+ infiltration but positively correlated with Treg infiltration. Results are determined by the Spearman correlation. Source data are provided as a Source data file.

including p53, cell cycle, E2F, and RNA polymerase II transcription pathways (Fig. 3b and Supplementary Table 2). A total of 58 proteins were identified both as Ppp2r1a-interacting proteins and *Ppp2r1a*$^{-/-}$ intestinal tumour organoid culture-enriched genes (Supplementary Table 3 and Fig. 3c). Furthermore, regulation of TP53 activity (*P* value = $4.10 \times 10^{-8}$), cell cycle (*P* value = $1.02 \times 10^{-6}$), and RNA polymerase II transcription (*P* value = $4.06 \times$

$10^{-6}$) were listed as the top candidates for the identified common gene sets. We identified four genes that were in common in the three gene sets, namely HDAC2, PPP2CB, AKT1, and Rb1, and these were further selected for expression and functional validations (Fig. 3d, e). Among them, the phosphorylation levels of both Rb and HDAC2 increased (Supplementary Fig. 3d). The former exhibited a decrease in the total protein level and the latter exhibited

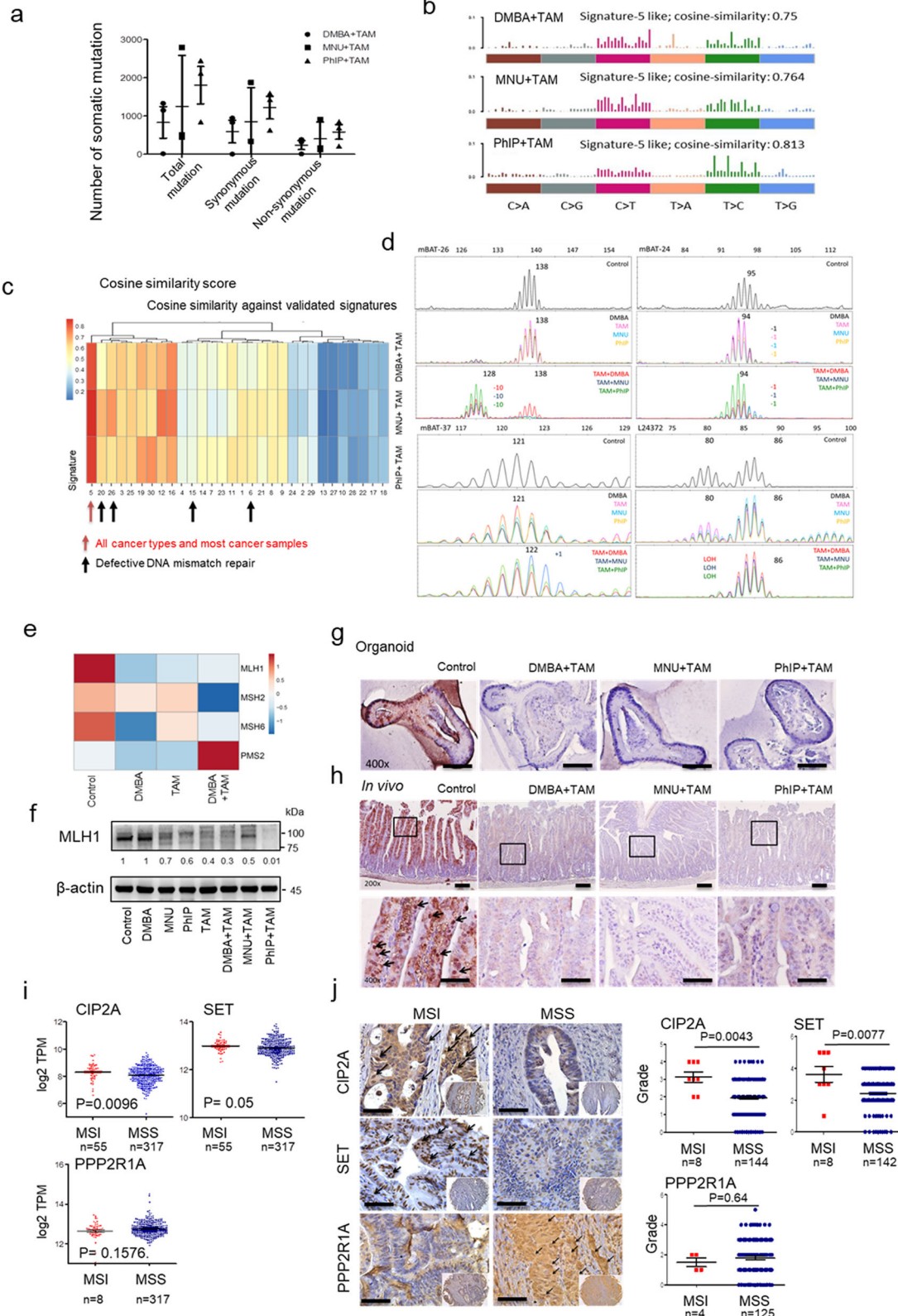

an increase in the total protein level (Supplementary Fig. 3d) in *Ppp2r1a*$^{-/-}$ intestinal tumour organoid cultures compared to the control. Moreover, E2F1 (Supplementary Fig. 3d), a Rb-interacting protein, and its downstream signalling pathway were also upregulated (Supplementary Table 2). Notably, Rb suppresses *dnmt3a/3b* promoter activity and expression by binding with the E2F1 protein to the *dnmt3a/3b* promoter[25], and HDAC2 plays an important role

in suppressing H3K9 acetylation (H3K9ac) in CRC[26]. As expected, we found that mouse *Ppp2r1a*$^{-/-}$ intestinal tumour samples showed higher levels of phospho-Rb and phospho-HDAC2 than the control (Supplementary Fig. 3e).

To further demonstrate that Rb and HDAC2 were the direct substrates of PP2A, CT26 cells were treated without and with LB100, a small-molecule inhibitor of PP2A[27], followed by

**Fig. 2 PP2A loss or inhibition is associated with MSI status. a–g** *Lgr5-EGFP-CreERT2; Ppp2r1a*$^{flox/flox}$ intestinal organoids were treated with 5 µg/ml DMBA, 50 µg/ml MNU, or 10 µg/ml PhIP in combination with or without tamoxifen (TAM) for 50 days. **a** Total numbers of somatic mutation events calculated from three biologically independent samples in each group display a very high mutational load. Data are denoted as mean ± s.e.m. **b** Mutational spectra of all base substitutions in organoid cultures treated with the indicated carcinogen and TAM. Similar mutation signatures were observed. **c** Heat map showing the cosine similarity scores for each indicated sample and COSMIC signature. The samples have been clustered according to the similarity score with each signature. The signatures have been ordered according to their similarity, such that very similar signatures cluster together. **d** The MSI status was evaluated by comparing mononucleotide repeats in each indicated sample. The mononucleotide regions mBAT-26, mBAT-37, mBAT-24, and L24372 were used to evaluate microsatellite instability. Mutant alleles are indicated with specific base pair as number(s) in each sample trace. Shifting or loss of heterozygosity (LOH) comparing to Control are marked with colour corresponding to each sample respectively. **e** Heat map of RNA-seq analysis of genes associated with DNA mismatch repair. **f** Western blotting and **g** immunohistochemical studies of MLH1 levels in organoid cultures treated with the indicated conditions. Blots are representative of two biological independent samples for each group. Images are representative of three biological independent samples for each group. Bar = 25 µm. **h** Ppp2r1a-loss colon tumours were induced in *Lgr5-EGFP-CreERT2; Ppp2r1a*$^{flox/flox}$ by treatment with DMBA, MNU or PhIP, and TAM for 36 days. Control colon tissues (Control) and Ppp2r1a-loss colon tumours were harvested for MLH1 level analysis by immunohistochemistry. Arrows indicate positive signals. Images are representative of three biological independent samples for each group. Bar = 25 µm. **i** Analysis of mRNA level expression were generated by the software from Gepia (http://gepia.cancer-pku.cn) using the data from TCGA and presented as mean ± s.e.m., revealing higher CIP2A ($n = 372$, $P = 0.0096$) and SET ($n = 372$, $P = 0.05$) and lower PPP2R1A ($n = 372$, $P = 0.1576$) levels in MSI compared to MSS. $P$ value was determined by two-sided unpaired $t$ test (CIP2A, SET) and Mann–Whitney $U$-test (PPP2R1A). **j** Human tissue array containing MSI and MSS colorectal tumours was assayed for CIP2A ($n = 152$, $P = 0.0043$), SET ($n = 150$, $P = 0.0077$) and PPP2R1A ($n = 129$, $P = 0.64$) levels by immunohistochemistry. (Left) Representative images are shown. Arrows indicate positive signals. Bar = 25 µm. (Right) Quantitative data are shown. $P$ value was determined by a two-sided Mann–Whitney $U$-test. Source data are provided as a Source data file.

Ppp2r1a pulldown of the cell lysates for an in vitro phosphatase assay and western blotting. The data showed that the Ppp2r1a pulldown in LB100-treated cell lysate exhibited decreased PP2A activity (Supplementary Fig. 4a) and increased pRb and pHDAC2 levels (Supplementary Fig. 4b). The biochemical evidence shows that HDAC2 and Rb are direct PP2A substrates. We have also demonstrated the involvement of PP2A-Rb-E2F in upregulating DNMT3A/3B and the involvement of PP2A-HDAC2 in suppressing H3K9ac to induce epigenetic silencing of *MLH1*, we genetically modified a mouse MSS CRC cell line, CT26. CT26 transfected with two short hairpin RNAs (shRNAs) against Ppp2r1a exhibited increased phospho-Rb, E2F1, DNMT3A/3B, and total and phospho-HDAC2 levels; decreased Rb, H3K9ac, and MLH1 levels (Fig. 3f); and induced an MSI state (Supplementary Fig. 5), compared to cells transfected with control shRNAs. Moreover, Rb knockdown transiently increased E2F1 and DNMT3A/3B levels but decreased MLH1 levels (Supplementary Fig. 6a), consistent with the fact that passive DNA methylation only suppresses gene expression transiently[28]. Interestingly, the effects of Ppp2r1a or Rb knockdown were also observed in a mouse MSI CRC cell line, MC38, in which the suppression of MSH2 was more pronounced than that of MLH1 (Supplementary Fig. 6b). More importantly, these Ppp2r1a knockdown effects were almost completely abrogated by E2F1 (Fig. 3g) or HDAC2 knockdown (Fig. 3g and Supplementary Fig. 6c) or treatment with a DNMT inhibitor, 5 aza-cytidine (Fig. 3g). Together, these data suggest the requirement of both Rb-E2F-DNMT3A/3B and HDAC2-H3K9ac for PP2A inactivation-mediated epigenetic silencing of *MLH1* or *MSH2* genes and induction of MSI status. As expected, we found that human colorectal tumour samples showed higher levels of phospho-Rb and phospho-HDAC2 than the control (Fig. 3h). Together, these data indicate that Rb and HDAC2 are dephosphorylated by PP2A and that their phosphorylation caused by PP2A inactivation for MLH1 epigenetic silencing can induce MSI status in CRC.

**PP2A inactivation induces ICB response.** Therefore, we assessed the effect of Ppp2r1a knockdown on tumour growth and response to antibodies against PD-1 (anti-PD1) in the MSS CRC cell line, CT26 (Fig. 4a). Ppp2r1a knockdown did not affect CT26 growth in syngeneic animals in the presence of control antibodies (Fig. 4b), however inhibited tumour growth in the presence of anti-PD1, compared to cells transfected with control shRNAs (WT CT26) (Fig. 4c). Although WT CT26 did not respond to anti-PD1 treatment (Fig. 4d), CT26 with Ppp2r1a knockdown responded to anti-PD1 treatment (Fig. 4e), suggesting Ppp2r1a knockdown sensitises CT26 to anti-PD1 treatment (Fig. 4e). Increased levels of CD8+ tumour-infiltrating T cells were found in tumours formed by CT26 cells with Ppp2r1a knockdown compared to those formed by WT CT26 cells (Fig. 4f). To demonstrate that Ppp2r1a knockdown converted cold tumours into hot tumours by increasing neoantigen, we submitted the RNA-seq data of CT26-shppp2r1a and CT26-scr tumour samples, integrated in the fastq.gz files, and applied the NAP-CNB[29] to predict neoantigens. A total of 270 missense transcripts, corresponding to 220 genes, shared by three CT26-shppp2r1a tumours but not found in the CT26-scr tumour were identified (Fig. 4g). The software also generated a ranking of putative neoantigens that are common in the three CT26-shppp2r1a tumours samples. The 30 top-scoring putative neoepitopes are shown in Supplementary Table 5. We then analysed the repertoire of T cell receptor (TCR) rearrangements in mice that received either WT CT26 or CT26 with Ppp2r1a knockdown tumour cells. To determine productive TCR rearrangements (TCRβ complementary determining region 3), DNA from peripheral blood mononuclear cells was amplified with TCR-specific primers and subjected to next-generation sequencing (NGS). Bioinformatics analyses revealed an expansion of the 20 more-represented TCR rearrangements in mice receiving CT26 with Ppp2r1a knockdown tumour cells (Fig. 4h and Supplementary Table 6), suggesting that Ppp2r1a knockdown in mouse MSS CRC triggers neoantigen generation and sensitises them to ICB-mediated anti-tumour activity.

In rodent models, direct deletion of MMR genes, such as *MLH1*[30] or *MSH2*[31], in tumours triggers neoantigen generation and cytotoxic T cell infiltration, impairs tumour growth, and sensitises them to ICB therapies. We reasoned that pharmacological agents that drive PP2A inactivation and cause epigenetic silencing of MMR genes may, paradoxically, be beneficial for therapeutic purposes. The safety, tolerability, and preliminary anti-tumour activity of LB100 have been previously shown in adult patients with progressive solid tumours[27]. Treatment of CT26, MC38, mouse triple-negative breast cancer 4T1, and mouse pancreatic cancer Pan18 with LB100 induced MLH1 loss and an increase in MSI status (Supplementary Fig. 7). We then

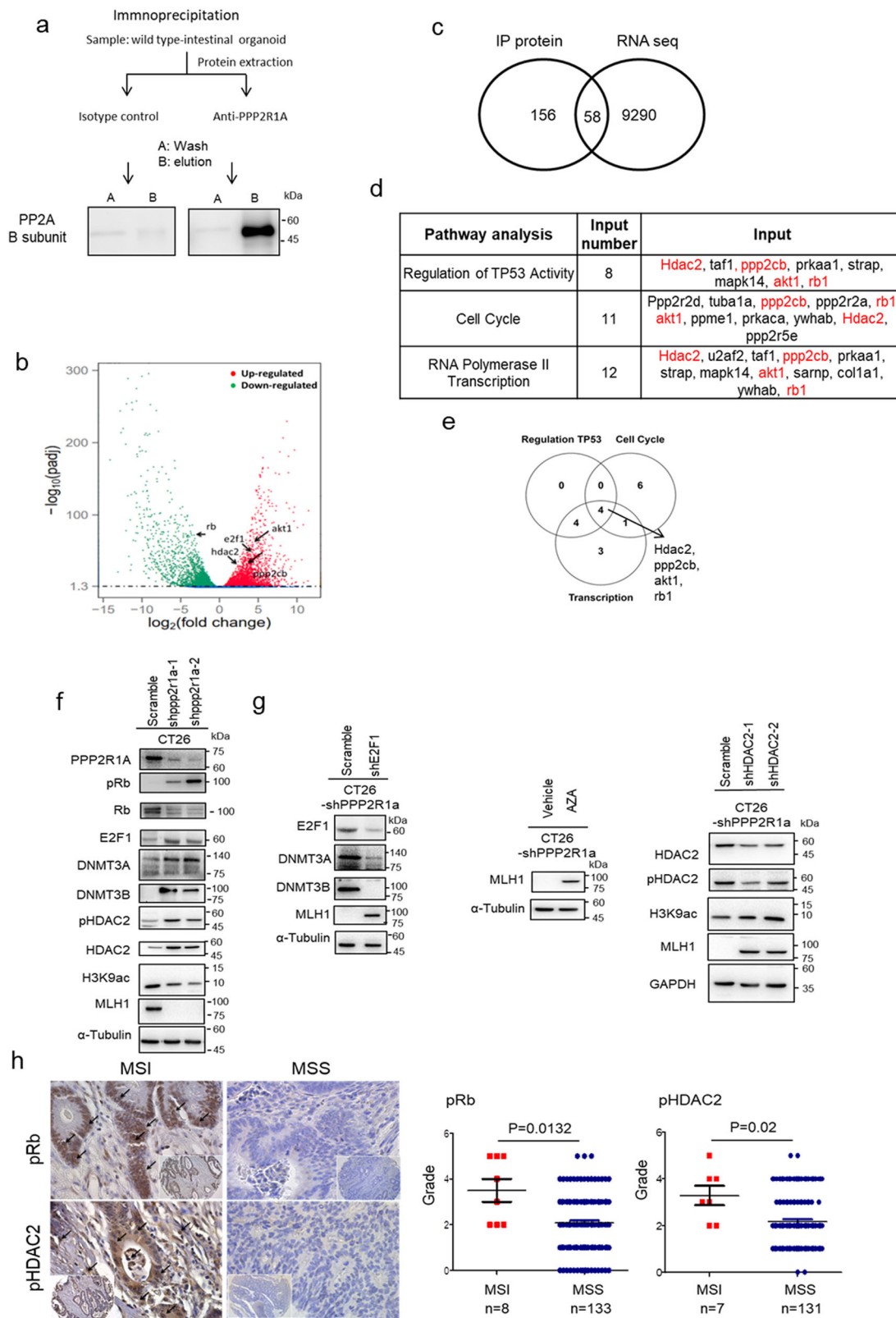

assessed the therapeutic effects of LB100 combined with anti-PD-1 treatment on tumour growth. Notably, pharmacologic inhibition of PP2A with LB100 has been known to enhance the immune-mediated anti-tumour activity of the PD-1 blockade through the inhibition of Treg differentiation[32]. Compared with treatment with LB100 or anti-PD1 alone, the combination of LB100 and anti-PD1 synergistically improved its activity on anti-

tumour growth and increased survival (Fig. 5a), although the synergistic effect of LB100 and anti-PD1 was less than that of Ppp2r1a knockdown and anti-PD1 (Fig. 4c, e). To prove that LB100 sensitised tumour cells to ICB therapies regardless of its Treg inhibitory activity, we showed the expression of p110δ in Treg (Foxp3+), but not in CD8+ or polymorphonuclear myeloid-derived suppressor cell (PMN-MDSC) (Ly6G^high) in

**Fig. 3 MLH1 loss caused by DNMT3A/B upregulation is mediated through the PP2A-Rb-E2F pathway. a** Immunoprecipitation performed with Capturem Protein A columns. Organoid lysates from WT mice were incubated with 1 µg of PPP2R1A antibody for 10 min. The antibody–lysate complex was applied to equilibrate Protein A spin columns. The eluted fraction was then subjected to SDS-PAGE to confirm the presence of PP2A B subunit antigen (52 kDa, detected by Cell Signaling #2290). Images are representative of one biological independent sample for each group. **b** Volcano plot for the comparison of RNA-seq data of *Lgr5-EGFP-CreERT2; Ppp2r1a*$^{flox/flox}$ intestinal organoids treated with and without DMBA and tamoxifen for 50 days. Differently expressed genes (fold change >2 and FDR < 0.05) are denoted in red or green. Plot is representative of three biological independent samples for each group. **c** Intersection of ppp2r1a interaction protein in **a** and differently expressed genes in **b** data (IP protein data are available in Supplementary Data 1 and RNA-seq data are available in GSE120241 with NIH/NCBI at GEO dataset). **d** The top 3 significant gene ontology categories from the 58 interacted proteins identified by GSEA (whole gene ontology category is listed in Supplementary Table 4). **e** Venn diagram showing overlap (gene numbers) among genes associated with regulation of TP53 activity, cell cycle, and RNA polymerase II transcription. The intersection contained four genes, including hdac2, ppp2cb, akt1, and rb1. Western blot analysis of **f** CT26 transfected with the indicated shRNAs, and treated with **g** vehicle control or 5-azacytidine (AZA, 1 µM for 1 days). Blots are representative of three biological independent samples for each group. **h** Human tissue array containing MSI and MSS colorectal tumours was assayed for p-Rb (n = 141, P = 0.0132) and p-HDAC2 (n = 138, P = 0.02) levels by immunohistochemistry. (Left) Representative pictures showing increased p-Rb and p-HDAC2 levels in MSI compared to MSS. Arrows indicate positive signals. Bar = 25 µm. (Right) Quantitative data are shown. P value was determined by two-sided Mann–Whitney U-test. Source data are provided as a Source data file.

the CT26 tumour microenvironment (Supplementary Fig. 8). We then used the p110δ inhibitor PI-3065 to block Treg-mediated immune suppression in mice[23], and showed that the therapeutic effects of the combination of LB100 and anti-PD1 on reducing tumour growth and enhancing survival were also observed in the presence of PI-3065 (Fig. 5b). Together, these data indicate that LB100 sensitises tumour cells to ICB therapies independent of its Treg inhibition activity.

To assess whether the results obtained using mouse cancer models can translate to human diseases, we characterised and compared the expression levels of proteins related to PP2A structure and functions between MSS and MSI tumours in 33 human CRC cell models. Compared to MSS cancers, MSI cancers had increased expression of CIP2A and decreased expression of PPP2R1A (Fig. 5c). We chose 3 human MSS CRC cells, HT29, SW480 (primary), and SW620 (metastasis) for further studies. The latter two were from the same patient[33]. Treatment of HT29 with pharmaceutical inhibition of PP2A activity (Supplementary Fig. 9) with two drugs, LB100 and LB102, for 2 and 7 days or with shRNAs against PPP2R1A induced MLH1 loss (Fig. 5d) and an increase in MSI status (Fig. 5e, Supplementary Fig. 10). Similar results were observed for SW620 (Fig. 5e, Supplementary Fig. 11). Moreover, treatment with LB100 also induced Rb phosphorylation and DNMT3A/3B upregulation in SW620 and SW480 (Supplementary Fig. 12). In line with the data obtained from mouse models, human CRC cells, in which MSI status is related to increased CIP2A level and decreased PPP2R1A level and in which the loss of MMR genes and the induction of MSI status are caused by PP2A inhibition, may also respond to ICB therapy when it is combined with PP2A inactivation.

**PP2A-related gene mutations or expression changes can predict mutation burden, MMR status, and the response to ICB**. A previous study demonstrated that the MMR status can predict the clinical benefit of ICB across 12 tumour types[5], making ICB the first Food and Drug Administration-approved tissue-agnostic therapy for MSI-H cancers[34]. This result urges us to determine whether these findings can be extended to various clinical settings. Besides CRC, endometrial carcinomas are classified into MSI and MSS tumours[35]. Analysis of human endometrial carcinoma TCGA data (n = 373)[35], including 13, 4, and 4% patients with PPP2R1A, SET, and CIP2A mutations or altered mRNA levels, respectively, revealed that patients with these genetic or transcriptional alterations had increased mutation rate cluster, MSI status, and tumour mutation count (Supplementary Fig. 13). We have further undertaken a reanalysis of the MSK-IMPACT cohort (including the clinical and genomic data of 1661 advanced cancer patients treated with ICB)[36] and showed that PPP2R1A

mutation (1.4%) was associated with an increased tumour mutation burden score and mutation count, and a better overall survival status (Supplementary Fig. 14a, b). Moreover, the median survival time and the univariate Cox regression hazard ratio of patients with PPP2R1A-mutated tumours were much better than those of patients with PPP2R1A-non-mutated tumours (Supplementary Fig. 14c). The pan-cancer nature of this biomarker probably reflects the fundamental mechanisms by which ICB functions. These data together support the hypothesis that PPP2R1A, SET, and CIP2A mutations or altered mRNA levels are associated with higher mutation burden and MSI status and help to predict responses to ICB.

## Discussion

Cancer immunotherapy with checkpoint-blocking antibodies targeting Cytotoxic T lymphocyte-associated antigen-4 (CTLA-4) and PD-1/PD-L1 can cause a long-term sustained response in patients with metastatic cancer of a wide range of histologies. Currently available immunotherapeutic agents are expensive and generally associated with considerable toxicity, and most importantly, only a small portion of cancer patients respond to immunotherapy, which mainly depends on the tumour-intrinsic and -extrinsic factors[37,38]. The identification of reliable predictive biomarkers[39] and the development of new combination therapies that enhance immunotherapy are the keys to successful treatment[40,41]. In the current study, we demonstrate that the loss of PP2A function through genetic manipulation or drug treatment makes a variety of mouse immune tolerance tumour models sensitive to ICB, and it has been further confirmed by using human cancer cell lines and human CRC tissue arrays. In addition, we show that human endometrial carcinomas with PPP2R1A, SET and CIP2A mutations or changes in mRNA levels are associated with higher mutation burden and MSI status.

Combinational therapy can be used to change the intrinsic or extrinsic factors of the tumour, thereby increasing the success rate of anti-cancer immunotherapy. For example, a compound (D18) can be used to increase KDM5A level for PD-L1 upregulating through PI3K-AKT-S6K1 signalling, and to activate Toll-like receptors 7 and 8 (TLR7/8) signalling pathways[40], thereby enhancing anti–PD-1 immunotherapy response in melanoma. Otherwise, an immunotherapeutic combination using the STAT1-activating IFN-γ, the TLR3 ligand poly(I:C), and an anti-IL-10 antibody can be used to convert the microenvironment to a more favourable configuration and sensitise murine tumours to ICB by attracting IFN-γ -producing nature killer (NK) cells[41]. These studies imply that it is possible to make the pa'ient's tumour sensitive to ICB through combinational therapies.

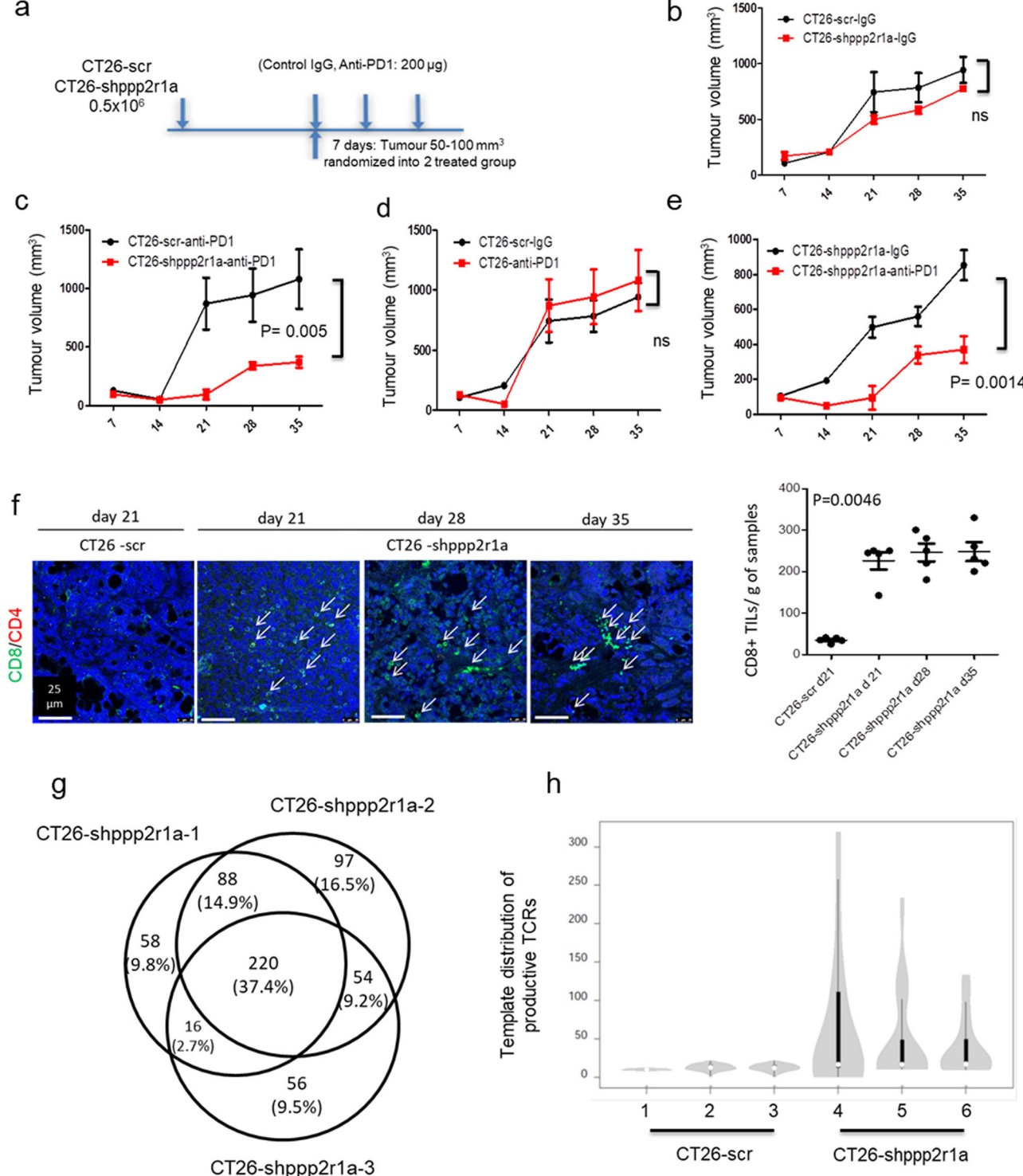

Similarly, the negative effects of PP2A inhibition in Treg function have been used to modify the immune microenvironment[32,42], thereby enhancing the response to ICB in murine tumour models. Taffs et al.[43]. was the first to identify PP2A as a potential negative regulator of T-cell activation through regulation of TCR mediated transmembrane signalling. Using okadaic acid, they proved that PP2A inhibition can enhance the cell-cell contact dependence of lymphocytes and antigen-specific effector functions. This result was later corroborated by Parry et al.[44], in which they identified PP2A as the phosphatase responsible for CTLA-4 mediated deactivation of

Akt signalling, providing direct evidence that PP2A plays a prominent role in mediating CTLA-4 suppression of T-cell activation. In addition, PP2A activity was also found to be elevated in Tregs compared to conventional T cells through a Foxp3-involved ceramide signalling pathway[42]. Treg-specific genetic deletion of PP2A increased mTORC1 signalling, and strikingly, resulted in Treg dysfunction, impaired immunosuppressive capability, and profound multiorgan autoimmune diseases[45]. A recent study further showed that PP2A inhibition with LB-100 enhanced the responses of murine tumours to ICB, potentially through activation of mTORC1 which resulted in reduced

**Fig. 4 Ppp2r1a knockdown triggers neoantigen generation and sensitises to ICB therapies in MSS tumours.** Ppp2r1a knockdown and PD-1 blockade synergistically elicit tumour rejection in a CD8+ T cell-dependent manner. **a** The flowchart of animal experiments. BALB/c mice were inoculated with 0.5 × 10[6] CT26 cells transfected with scramble (scr) or Ppp2r1a shRNA (shppp2r1a) subcutaneously in the flank. In all experiments, polyclonal hamster IgG and Anti-PD-1 (clone: RMP1-14, BioXcell) were administered using doses of 200 μg for the initial injection and 100 μg for subsequent injections. Mice were then intraperitoneally injected with isotype control IgG or anti-PD1 antibodies (200 μg, per mouse) on days 5, 8, and 11 following tumour injection. **b–e** The indicated CT26 clones were injected into syngeneic BALB/c mice. Tumour growth curves were compared and tumour samples were harvested for further analysis. Three independent experiments were done. Data of representative experiments are shown as mean ± s.e.m. (n = 5). P value was determined by two-way ANOVA. **f** (Left) Immunofluorescence of CD8 and CD4 was performed on CT26-scr and CT26-shppp2r1a tumours to assess CD8+ and CD4+ T cell infiltration at the indicated time points. Bar = 25 μm. **f** (Right) Quantification of CD8+ density. Three independent experiments were done. Data of representative experiments are shown as mean ± s.e.m. (n = 5). P = 0.0046 was determined by two-sided one-way ANOVA. **g** A Venn diagram showing a total of 220 genes, including 270 missense transcripts, shared by three CT26-shppp2r1a tumours but not found in the CT26-scr tumour. **h** Distribution of the 20 most frequent TCR rearrangements identified in peripheral blood from mice (n = 3 for each group) injected with the indicated CT26 clones. The width of the violins is proportional to the number of TCR templates in each level of the y axis; the bars inside the violins show the quartiles of the 20 templates; bars span the first to third quartiles; the horizontal lines inside the bars represent the median; the grey zones represent all samples; the white circle shows the median value. TCR analysis was performed on blood samples obtained 13 days after injection of the tumour cells, as described in the "Methods" section. Source data are provided as a Source data file.

differentiation toward Tregs and increased tumour infiltrating CD8+ T cells[32]. Together, these data suggest that PP2A inhibition can enhance the anti-tumour response of the adaptive immune system by inducing Treg dysfunction.

The current study showed that pharmaceutical inhibition of PP2A with LB-100 in different murine tumour models induced MMR gene silencing and a MSI status, triggered neoantigen production, and enhanced infiltration of cytotoxic CD8+ lymphocytes into tumours, thereby synergistically improving their activity against tumour growth and enhancing survival. In addition, simultaneous treatment with the Treg inhibitor PI-3065 did not reduce their synergistic effect in reducing tumour growth and improving survival, indicating that LB-100 also sensitised tumour cells to ICB in a Treg-independent manner. Combining current and previous studies, PP2A inhibition can sensitise the ICB response by turning MSS cold tumours into MSI inflammatory tumours, and enhance the ICB response through Treg inactivation. Together, these data indicate that PP2A inhibition can be used to change the intrinsic and extrinsic factors of tumours to improve the success rate of anti-cancer immunotherapy.

The small molecule LB-100 is one of the best-studied PP2A inhibitors so far. It has no obvious toxicity to animals, and the first human clinical trial has achieved encouraging results[27]. Since targeting PP2A can enhance ICB response through modification of both tumours-intrinsic and extrinsic factors, the in vivo anti-cancer immunotherapy effect of systemic administration of LB-100 should be better than that of Ppp2r1a gene knockdown in mouse cancer cells. However, we have observed that the method of using shRNAs against Ppp2r1a in cancer cell lines had better anti-cancer effects on mouse tumours models than the method of systemic administration of LB100. We speculate on the reasons why systemic administration of LB-100 lost its better effect. Although targeting Treg has been proposed as a combinational therapy for enhancing ICB response, however, the clinical trial results do not support this strategy. One such mechanism is the production of tryptophan metabolites along the kynurenine pathway by the enzyme indoleamine 2,3-dioxygenase 1 (IDO1), which is induced by IFNγ[46]. However, clinical trials using inhibition of IDO1 in combination with blockade of the PD1 pathway in patients with melanoma did not improve the efficacy of treatment compared to PD1 pathway blockade alone[47,48]. These data indicate that the previously proposed LB-100/anti-PD-1 combination for suppressing Treg to enhance ICB response may be ineffective[32].

Based on the consensus molecular subtypes (CMSs) of CRC[49], the majority of MSI tumours belong to type I CMS (characterised by increased expression of genes associated with a diffuse

immune infiltrate, mainly composed of TH1 and cytotoxic T cells), while tumours caused by $Apc^{fl/fl}$, $Kras^{LSL-G12D}$, $Tgfbr2^{fl/fl}$, and $Trp53^{fl/fl}$, belong to type IV CMS (characterised by TGFβ-activated stroma)[50] are corresponding to MSS, suggesting the tumour itself is the dominant force shaping the tumours microenvironment. The current study also supports this theory, in which murine ppp2r1a-deficient tumours exhibit enhanced CD8+ T cell infiltration and reduced Foxp3+ Treg infiltration. Similarly, higher SET and CIP2A levels in human CRC cancers are positively correlated with CD8+ cytotoxic T infiltration but negatively correlated with Foxp3+ Treg infiltration. Therefore, MSI CRC has the tumour microenvironment of type I CMS and is sensitive to ICB treatment, so there is no need to use Treg inhibition in combination. Because LB100, a PP2A inhibitor, has the ability to convert cold MSS tumours into hot MSI tumours, thereby inhibiting Treg infiltration by the tumour itself, there is no need to combine Treg inhibition for enhancing ICB response. We further showed that the effect of LB100 combined with anti-PD1 on the enhancement of ICB response was independent of Treg inhibition. Together, these data indicate that the effect of the LB-100/anti-PD-1 combination on enhancing ICB response is mainly through the conversion of MSS tumours into MSI tumours. This can also be used to explain why the in vivo anti-cancer effect of LB-100/anti-PD1 was worse than that of anti-PD1 combined with Ppp2r1a gene knockdown in mouse cancer cells.

Furthermore, the inability of the therapeutic agent to reach the tumour site may be the reason for the lower efficacy of systemic administration, which can be improved by using a cancer-targeting strategy combined with nanomedicine[51] that offers multiple benefits in treating chronic human diseases, including cancer, by site-specific and target-oriented delivery of medicines. High local drug concentration would be able to increase the exposure time, thereby enhancing the anti-cancer efficacy and reducing the systemic toxicity in the treatment of a variety of cancers[52]. Compared with the direct use of shRNAs against Ppp2r1a in cancer cell lines, the reduced anti-cancer effect observed in systemic administration of LB-100 may be attributed to the lower concentration of LB-100 in the tumour site, which should be improved by controlled delivery of therapeutic agents to tumours. To be noted, inhibiting PP2A, a tumour suppressor, can cause lots of oncogenic signals in normal tissues, thereby limiting the therapeutic potential of PP2A inhibition in cancer treatment. Similarly, this problem can be solved by controlled delivery of therapeutic agents to tumours.

Although it has been reported that some subunits of the PP2A holoenzyme, such as PPP2R2B[53], are highly methylated and silenced in some CRC tumours, PP2A inhibition may not induce

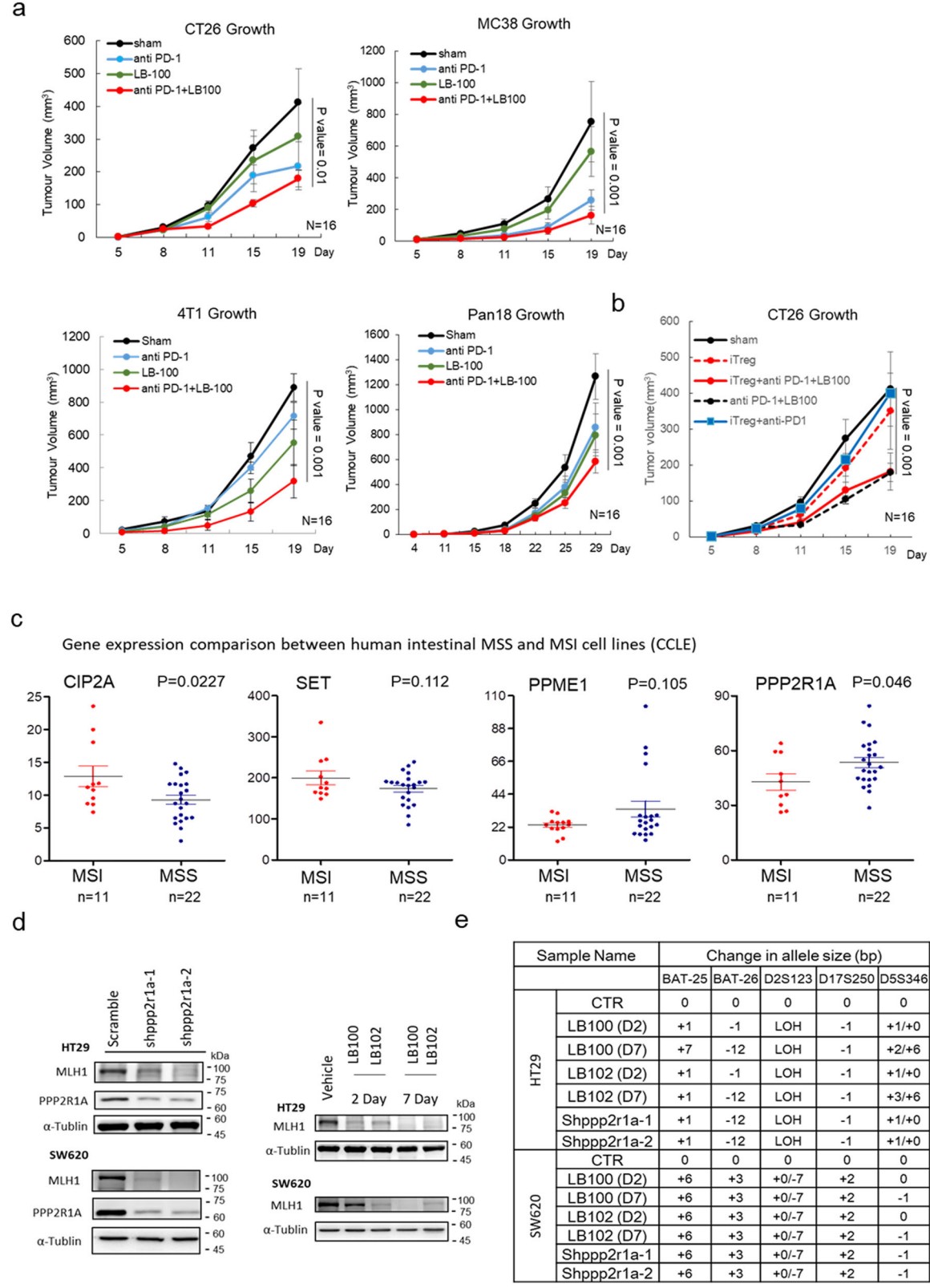

MSI status in these tumours. Due to intratumoural heterogeneity of genetic or epigenetic events leading to PP2A inactivation, some tumour cells may retain functional PP2A activity. For these PP2A functional tumour cells, PP2A inhibition may induce MSI status, thereby making them sensitive to ICB.

There are limitations to our study. We only proved the effect of PP2A inhibition in the "primary" resistance to ICB, reprograming

cold into hot inflamed tumours and improving anti–PD-1 immunotherapy, but did not examine whether mRNA dysregulation or mutations of these genes also play a role in the adaptive and acquired resistance to ICB[37], future studies will also need to address the unravelling complexity associated with these dynamic treatment responses[54]. Generally, applying the results of animal models to human patients is a key biomedical challenge. This

**Fig. 5 Combinational effect of PP2A inhibition and ICB on killing MSS tumours is independent of Treg inhibition. a** PP2A inhibition and PD-1 blockade synergistically elicit tumour rejection. BALB/c mice were inoculated with $0.5 \times 10^6$ CT26 or 4T1 cells, and C57BL/6 mice were inoculated with $0.5 \times 10^6$ MC38 or $0.5 \times 10^5$ Pan18 cells subcutaneously in the flank. When tumours reached between 50 and 100 mm$^3$, mice were randomised into four treatment groups, including Sham, anti-PD-1 (200 µg per mouse, clone: RMP1-14, BioXcell), LB100 (0.16 mg/kg), and anti-PD-1 with LB100 combination. Treatments were administered by intraperitoneal injection on days 5, 8, and 11. Three independent experiments were done. Data of representative experiments are shown as mean ± s.e.m. ($n = 16$). $P$ value was determined by two-sided two-way ANOVA. **b** For investigation of PP2A inhibition and PD-1 blockade synergistically eliciting tumour rejection in a regulatory T cell-independent manner, BALB/c mice inoculated with $0.5 \times 10^6$ CT26 subcutaneously in the flank were further randomised into mice treated with vehicle or with Treg inhibitor (iTreg), PI-3065 (75 mg/kg, daily), after tumour reached between 50 and 100 mm$^3$. Three independent experiments were done. Data of representative experiments are shown as mean ± s.e.m. ($n = 16$). $P$ value was determined by two-sided two-way ANOVA. Coefficient of drug interaction (CDI) was calculated and CDI < 0.7 indicates significant synergism. **c** Analysis of mRNA level data in human colorectal cancer cells lines, extracted from the Cancer Cell Line Encyclopedia (CCLE), revealed higher CIP2A and SET levels and lower PPP2R1A in MSI compared to MSS ($n = 33$ for each gene expression). $P$ value was determined by a two-sided unpaired $t$ test. **d** Western blot analysis and **e** MSI status evaluation of HT29 and SW620 treated with vehicle control (CTR), 2.5 µM LB100, and 2.5 µM LB102 treatment for 2 days (D2) and 7 days (D7) or treated with different Shppp2r1a. Blots are representative of two biological independent samples for each group. The mononucleotide regions BAT-25, BAT-26, D2S123, D17S250, and D5S346 were used to evaluate microsatellite instability. Shiftings and loss of heterozygosity (LOH) comparing to control (CTR) are shown. Source data are provided as a Source data file.

problem is the main reason for the failure of treatment from preclinical research to clinical trial[55]. This is likely related to the heterogeneity of patients, cancer types, and environmental factors. However, these are all controlled in the mouse model and this reason can be used to explain the failure.

## Methods

**Animal studies**. *Ppp2r1a*$^{flox/flox}$ mice, carrying conditional alleles with *loxP* sites flanking exon 5-6 of *Ppp2r1a*, were purchased from the Jackson Laboratory and crossed to *Lgr5-EGFP-CreERT2* mice[56] to generate *Lgr5-EGFP-CreERT2;Ppp2r1a*$^{flox/flox}$. BALB/c and C57BL/6 mice were purchased from National Laboratory Animal Center (Taiwan). All animal studies and care of live animals were approved and performed following the guidelines made by the China Medical University Institutional Animal Care and Use Committee (No. 2017-239 and No. 2020-229). Mice aged 6–8 weeks were injected intraperitoneally with a single 200 µl dose of tamoxifen in sunflower oil at 10 mg/ml. Mice were housed in an animal facility at China Medical University Animal Center under constant environmental conditions (room temperature, 20–24 °C; relative humidity, 50–70%, and a 12-h light-dark cycle). All mice had access to food and water ad libitum.

**Mouse intestinal organoid cell isolation and culture**. The protocols of mouse intestinal organoid, cell isolation, and culture were modified from previously described methods[21,56]. In brief, the intestines were harvested, opened longitudinally, and cut into small pieces of 2 mm in size. The tissues were soaked in dissociation reagent and gently shaken at room temperature for 15 min, followed by filtering through a 70 µm sterile cell strainer. The crypts were then collected by centrifugation of the fluid mixture at $140 \times g$ for 5 min at 4 °C. An aliquots of 500 crypts was resuspended with 50 µl growth factor reduced phenol-free Matrigel (BD Biosciences, San Jose, CA, USA), reseeded, and polymerised in the centre well of a 24-well plate. The crypts were cultured in Dulbecco's modified Eagle's medium/F12 supplemented with penicillin/streptomycin, 50 ng/ml murine recombinant epidermal growth factor (Peprotech, Hamburg, Germany), Noggin (5% final volume), and R-spondin 1 (5% final volume). Medium change was performed every 3–4 days. The experiment was repeated twice, each condition with triplicate samples, and each sample contained multiple (>15) organoids.

**Tissue microarray (TMA) slides, immunohistochemistry, and immunofluorescence**. TMAs were collected from China Medical University Hospital (CMUH) with approval by Research Ethics Committee (protocol no. CHUH105-REC2-073), which all patients were included after signing informed consent and from human CRC TMAs provided by Tristar Technology Group (Rockville, MD, USA). For the preparation of paraffin sections, the intestinal tissues and organoid cells were rinsed three times in ice-cold phosphate-buffered saline (PBS), followed by spin down at $150 \times g$ for 10 min at 4 °C and harvested for tissue fixation, and subsequent paraffin-embedding and sectioning (5 µm). Sections were deparaffinized and stained with H&E for histology analysis. For immunofluorescence, the paraffin sections were permeabilized with 0.3% Triton-X 100 in PBS for 20 min and incubated with primary antibodies overnight at 4 °C, followed by washes and incubation with corresponding secondary antibodies for 30 min at room temperature. The primary antibodies included anti-MLH1 (ab92312, dilution 1:500, Abcam), anti-CD3E (A1753, dilution 1:50, ABclonal), anti-CD4 (ab237722, dilution 1:100, Abcam), anti-CD8 (ab217344, dilution 1:500, Abcam), anti-CD20 (ab64088, dilution 1:100, Abcam) and anti-FOXP3 (GTX107737, dilution 1:100, GeneTex). The secondary antibodies used were DyLight® 650 Conjugated goat anti-rabbit (A120-101D5, Bethyl) and Goat anti-Rabbit IgG Antibody-FITC

(A120-201F, Bethyl). The slides were mounted with DAPI (GTX30920, GeneTex) followed by covering with a coverslip. The specimens were examined with a Leica TCS SP8X confocal spectral microscope (Leica Application Suite X). For immunohistochemical staining, paraffin-embedded sections were deparaffinized and rehydrated, with antigen being retrieved by placing sections in Declere working solution (Cell Marque, Austin, TX) in 95 °C water for 30 min. Endogenous peroxidase activity was blocked by 3% hydrogen peroxide. Residual enzymatic activity was removed by washes in PBS, and nonspecific staining was blocked with Ultra V Block for 5 min (Thermo Fisher Scientific, Fremont, CA). Then, the sections were reacted with primary antibodies anti-PPP2R1A (GTX102206, dilution 1:100, GeneTex), anti-pHDAC2 (GTX50236, dilution 1:50, GeneTex), anti-pRb (ab173289, dilution 1:100, Abcam), anti-CIP2A (ab99518, dilution 1:100, Abcam) and anti-SET (ab181990, dilution 1:250, Abcam) followed by incubation with corresponding biotinylated secondary antibodies (Vector Laboratories, Burlingame, CA), and then these sections were treated with streptavidin-peroxidase (LSAB Kit; Dako, Carpinteria, CA), followed by diaminobenzidine staining. Counterstaining was performed with Mayer's haematoxylin and photographed with Zeiss AxioImager Z1 microscope system (Wetzlar, Germany) and an automated acquisition system (TissueGnostics, Vienna, Austria). Pictures were acquired using the Tissue-Faxs software (TissueGnostics). The percentage of positively stained cells was determined using the HistoQuest software (TissueGnostics)[57].

**Transcriptome analysis**. Integrity of RNA extracted from organoid culture using an RNeasy Kit (Qiagen) was assessed using the RNA Nano6000 assay kit (Agilent Technologies, CA, USA). Library preparation and sequencing were performed using the Novogene Technology. The output data (FASTQ files) were mapped to the target genome (TopHat v2.0.12), and the splice junction database was generated according to the gene model annotation file. HTSeq v0.6.1 was used to count the reads numbers mapped to each gene. Then the FPKM of each gene was calculated according to the length of the gene and read counts mapped to this gene. ClustVis free web server[58] and GSEA[59] were used to analyse differential expression heat map results and biological variability, respectively. Data were submitted and approved by Gene Expression Omnibus (GEO; accession number GSE120241).

The TCGA RNA-seq data was downloaded from TCGA repositories within dbGAP (now part of the Cancer Genomics Cloud), followed by alignment of the RNA-seq data to the genome (hg19) using Mapsplice and quantified relative to the transcriptome using RNA-seq by expectation–maximisation (RSEM).

The TIMER database, including 10,897 samples across 32 cancer types from TCGA, is a comprehensive resource for systematic analysis of immune infiltrates across diverse cancer types (https://cistrome.shinyapps.io/timer/)[60]. We analysed CIP2A, SET, PPME-1, and PPP2R1A expression in CRC and the correlation with the abundance of immune infiltrates, including B cells, CD4+ T cells, CD8+ T cells, neutrophils, macrophages, and dendritic cells, via gene modules. The gene markers of tumour-infiltrating immune cells included markers of CD8+ T cells, T cells (general), B cells, monocytes, TAMs, M1 macrophages, M2 macrophages, neutrophils, natural killer (NK) cells, dendritic cells (DCs), T-helper 1 (Th1) cells, T-helper 2 (Th2) cells, follicular helper T (Tfh) cells, T-helper 17 (Th17) cells, Tregs, and exhausted T cells. The correlation module produced the expression scatter plots between a pair of user-defined genes in a certain cancer type, together with Spearman's correlation and the estimated statistical significance. For gene correlation analysis, GEPIA[61] was applied to perform the correlation between given sets of TCGA expression data. The Spearman method was used to determine the correlation coefficient.

**Cell lines**. CT26 mouse colon carcinoma, SW480, SW620 human colon carcinoma and 4T1 mammary carcinoma cell lines were obtained from ATCC. MC38 colon carcinoma was obtained from Kerafast. Mouse pancreatic ductal adenocarcinoma

(clone Pan18)[62] was generated from transgenic mice bearing pancreatic cancers with the following genotype: elastase-CreER; LSL-Kras$^{G12D}$; p53$^{+/-}$ mice. Pan18 cells were kindly provided by Dr. Chia-Ning Shen (Academia Sinica, Taiwan). All cell lines used were tested and shown to be negative for mycoplasma contamination using PCR amplification. HT29 and SW620 were cultured in the complete medium by RPMI 1640 and DMEM, respectively with 10% foetal bovine serum and 1% Penicillin–Streptomycin–Amphotericin B Solution (Biological Industries, USA) at 37 °C and 5% $CO_2$. For treatment with LB100 (CC7693, ChemCatch) and LB102 (a gift from the Medicinal chemistry team in New Drug Development Centre of China Medical University), HT29 or SW620 cells were resuspended in a serum-free medium and incubated for 4 h to avoid the sticking of drugs by proteases and albumins in serum, followed by incubation in complete medium without (vehicle control) or with 2.5 μM LB100 or 2.5 μM LB102 with medium change every 2 days, when vehicle, LB100, or LB102 were continuously added. Cells were then harvested, and cell pellets were used for western blotting and MSI status analysis at 2 (D2) and 7 days (D7). Control (vehicle) cells were analysed at 7 days.

**Whole-exome sequencing, alignment, and annotation**. SureSelectXT Mouse All Exon Kit (G7550E-001, Agilent, CA, USA) was used to capture exome sequences according to the standard protocols. The products of exome capture should meet the criteria of 300 ± 30 bp length of fragments and total amount >600 ng. After then, the index-tagged samples were pooled and sequenced on Illumina HiSeq 2000. Burrows-Wheeler Alignment (v0.7.12) was applied to align the reads to the reference genome (mm10) with default parameters, which were sorted and the duplicated reads were marked by picard-tools (v1.8). Indel realignment and base quality score recalibration were performed with GenomeAnalysisTK (v3.5) using mm10 dbsnp database as known sites. Single-nucleotide polymorphisms and indels were identified by GenomeAnalysisTK HaplotypeCaller (v3.5) with default parameters. Whole-exome sequencing raw data were submitted to SRA database (SRA; http://trace.ncbi.nlm.nih.gov/Traces/sra/, accession number SRP162613)

**MSI analysis**. MSI in mouse cells was determined using a panel of three microsatellite markers as previously described[63]. Amplification was performed with the following mouse sample primers: L24372, forward 5′-GGGAAGACTGCTTAGGG AAGA-3′ and reverse 5′-ATTTGGCTTTCAAGCATCCATA-3′; mBAT-24, forward 5′-CATAGACCCAGTGCTCATCTTCGT-3′ and reverse 5′-CATTCGGTG GAAAGCTCTGA-3′; mBAT-26, forward 5′-TCACCATCCATTGCACAGTT-3′ and reverse 5′-CTGCGAGAAGGTACTCACCC-3′; mBAT-37, forward 5′-TCTG CCCAAACGTGCTTAAT-3′ and reverse 5′-CCTGCCTGGGCTAAAATAGA-3′. For human CRC cells, primers used were: BAT 25, forward 5′-TCGGCTCCAAG AATGTAAGT-3′ and reverse 5′-TCTGCATTTTAACTATGGCTC-3′; BAT 26, forward 5′-TGACTACTTTTGACTTCAGCC-3′ and reverse 5′-AACCATTCAAC ATTTTTAACCC-3′; D17S250, forward 5′-GGAAGAATCAAATAGACAAT-3′ and reverse 5′-GCTGGCCATATATATATTTAAACC-3′; D2S123, forward 5′-AA ACAGGATGCCTGCCTTTA-3′ and reverse 5′-GGACTTTCCACCTATGGGA C-3′; D5S346, forward 5′-ACTCACTCTAGTGATAAATCG-3′ and reverse 5′-AG CAGATAAGACAGTATTACTAGTT-3′. The PCR reaction was performed in 20 μl of PCR reaction using Platinum Taq Polymerase Kit from Invitrogen and 20 ng of DNA. The cycling profile was as follows: 94 °C for 4 min; 94 °C for 30 s, 56 °C for 45 s and 72 °C for 30 s for a total of 35 cycles. A final extension was set at 72 °C for 6 min to complete the amplification. PCR fragments were separated on a 3730 DNA analyser (Applied Biosystems) and raw data were analysed with the GeneMapper software.

**Western blot analysis**. For western blot analysis, the organoids or cells were lysed in lysis buffer (1% Triton X-100, 150 mmol/l NaCl, 10 mmol/l Tris pH 7.4, 1 mmol/l EDTA pH 8.0, protease/phosphatase inhibitor cocktail, Bio Kit). Lysates were clarified by centrifugation and amounts of proteins were normalised with the Protein Assay Dye Reagent concentrate kit (Bio-Rad). Total cellular proteins were extracted by solubilizing the cells in boiling 4X laemmli sample buffer (Bio-Rad) and 2-mercaptoethanol (Sigma). Samples were boiled for 10 min and sonicated for 30 s. Samples were then separated on 10% Bis-Tris polyacrylamide gels and transferred to polyvinylidene difluoride (PVDF) membrane (Millipore). Western blot detection was performed with the enhanced chemiluminescence system (PerkinElmer) and peroxidase-conjugated secondary antibodies (Merck). The following primary antibodies were used: anti-PPP2R1A (GTX102206, dilution 1:500, GeneTex), anti-pHDAC2 (GTX50236, dilution 1:500, GeneTex), anti-HDAC2 (GTX01527, dilution 1:500, GeneTex), anti-H3K9ac (GTX88007, dilution 1:500, GeneTex), anti-pRb (8180, dilution 1:1000, CST), anti-Rb (9313, dilution 1:1000, CST), anti-E2F1 (sc-193, dilution 1:200, Santa Cruz), anti-DNMT3A (3598, dilution 1:1000, CST), anti-DNMT3B (GTX22851, blot by 2.0 μg/ml, GeneTex), anti-MLH1 (ab92312, dilution 1:2000, Abcam), anti-MSH2 (bs-0758, dilution 1:500, Bioss Antibodies), anti-MSH6 (bs-3804, dilution 1:300, Bioss Antibodies), anti-GAPDH (GTX100118, dilution 1:5000, GeneTex), anti-Actin (MA5-15739, dilution 1:1000, Invitrogen), and anti-Tubulin (GTX112141, dilution 1:500, GeneTex).

**Immunoprecipitation and mass spectrometry**. For the identification of PPP2R1A-interacting proteins, wild-type intestinal organoid cultures were lysed for protein immunoprecipitation with a PPP2R1A-specific antibody (GTX102206,

dilution 1:50, GeneTex) or an isotype IgG control antibody (401201, dilution 1:50, Biolegend) by using Capturem Protein A columns. Following by centrifugation, washing steps and elution was carried out in 100 μl of elution buffer. Fractions were resolved by gel electrophoresis, transferred onto PVDF membranes, and probed with an anti-PP2A B subunit antibody (2290, dilution 1:1000, CST). Eluates were trypsin digested and desalted on C18 ZipTips (Millipore) before analysis on a hybrid quadrupole-orbitrap mass spectrometer (Q Exactive; Thermo Fisher Scientific). Protein identifications were obtained with MASCOT (Matrix Science) using UniProt/Swiss-Prot[64]. The results were controlled for false-discovery rate and the proteins identified were analysed with Qiagen's Ingenuity Pathway Analysis (IPA) software for the identification of pathway enrichment in the data set.

**Transfection (shRNA construct, transfection)**. Cells were transfected with the shRNAs against PPP2R1A, Rb, HDAC2 or a negative control short hairpin RNA (shRNA) (Scramble) using Lipofectamine 2000 following manufacturer's recommendations (Invitrogen/Life Technologies, USA). After 48 h transfection, transfected cells were selected by puromycin antibiotics (2 μg/ml) for one week or until complete eradication of non-transfected control cells. Cells were resuspended and reseeded for removing dead cells. Cells were then harvested and cell pellets were used for western blotting or MSI status analysis. The shRNAs against mouse PPP2R1A (#1: TRCN0000012626 and #2: TRCN0000012624), mouse Rb (TRCN0000218613), mouse HDAC2(TRCN0000039397) and human PPP2R1A (#1: TRCN0000231600 and #2: TRCN0000231509) were purchased from the RNAi Core Facility (Academia Sinica, Taipei, Taiwan) and sequences are listed in Supplementary Table 1.

**TCR sequencing**. Mouse TCRβ complementary determining region 3 (CDR3) was amplified using the survey ImmunoSeq platform in a multiplex PCR method using 45 forward primers specific to TCR-Vβ gene segments and 13 reverse primers specific to TCR-Jβ gene segments (Adaptive Biotechnologies). RNA-seq libraries were prepared with the Illumina TruSeq RNA Sample Preparation kit and sequenced on an Illumina HiSeq4000 with single-end 50-bp reads, 8 samples per lane. Joint TCRα and β CDR3 repertoires were extracted from raw fastq files using MiXCR v2.1.11 (https://github.com/milaboratory/mixcr/releases/tag/v2.1.11).

**Neoantigen identification**. RNA-seq analyses were performed on CT26-shppp2r1a and CT26-scr tumours harvested from mice when tumour volume reached around 70 mm$^3$. Total RNA was isolated from tumours using the AllPrep DNA/RNA Mini Kit (Qiagen), according to the manufacturer's instructions. Prediction of neoantigens was performed by submission of RNA-seq data in "fastq.gz" files to generate a ranking of putative neoantigens via the NAP-CNB website[29]. For each variation, the mutant peptide sequence was obtained. Haplotypes for mouse samples were set to H2-Kd and H2-Dd for BALB/c background. Data were submitted and approved by Gene Expression Omnibus (GEO; accession number GSE182571).

**Bisulfite PCR sequencing**. Genomic DNA was isolated from mouse intestinal organoid cultures using QIAamp DNA Mini Kit (Qiagen). In all, 500 μg of DNA was treated with sodium bisulfite using EZ DNA Methylation-Gold Kit (ZYMO Research). The bisulfite converted DNA was amplified by PCR using Bio-Rad HotStar plus Taq DNA polymerase and the following primers are used: mMLH1 Primer I, 5′-GGTGTACGAAGTTATTTTTATTTTAGTC-3′ and mMLH1 Primer II: 5′-ACCCAACGATACCTAATAATAAAACC-3′. Cycling conditions were as following: denaturation at 95 °C for 5 min, followed by 45 cycles of amplification (95 °C for 30 s, annealing temperature for 30 s, and 72 °C for 30 s), and a final extension at 72 °C for 10 min. The PCR products were run on a 3% agarose gel and recovered from the gel using QIAQuick Gel Extraction Kit (Qiagen). The purified PCR products were submitted for sequencing (SeqWright), if their concentration was high enough or subjected to second round of PCR for further enrichment. On the chromatograms of sequencing results, the heights of methylated peak (C peak) and unmethylated peak (T peak) of each CpG were measured and the percentage of methylation was calculated as: methylation% = 100 × value of methylated peak/ (value of methylated peak + value of unmethylated peak). The overall DNA methylation of a gene promoter region was the average of methylation percentages of all CpGs examined.

**Methylation-specific PCR (MSP)**. MSP has been well established for measuring the status of specific CGI methylation of MLH1 genes in mouse[65]. MSP analysis of bisulfite-converted DNA was performed using methylated (M) and unmethylated (U) primer sets, which are Mlh1 Unmethylated DNA forward primer, 5′-TTTGTTGTAGATGTTTGATTAGGGTTGT-3′ and Unmethylated DNA reverse primer, 5′-CCACCTAATCACCTCTACTCAAAAACACA-3′; Methylated DNA forward primer, 5′-TCGTAGACGTTCGATTAGGGTCGC-3′ and Methylated DNA reverse primer, 5′-TAATCGCCTCTACTCGAAAACGCG-3′. The cycling conditions were the same as for Bisulfite PCR sequencing above. The PCR products were run on a 3% agarose gel.

**Tumour cell transplantation studies**. BALB/c mice were inoculated with $0.5 \times 10^6$ CT26 or 4T1 cells, and C57BL/6 mice were inoculated with $0.5 \times 10^6$ MC38 or $0.5 \times 10^5$ Pan18 cells subcutaneously in the flank. When tumours reached between 50 and 100 mm$^3$, mice were randomised into four treatment groups, including Sham, anti-PD-1 (clone: RMP1-14, BioXcell), LB100 (0.16 mg/kg), and anti-PD-1 with LB100 combination. Treatments were administered by intraperitoneal injection on days 5, 8, and 11. For anti-PD-1 administered doses, 200 µg per mouse was given for the initial treatment day and 100 µg per mouse was given for the subsequent treatment day.

For investigation of PP2A inhibition and PD-1 blockade synergistically eliciting tumour rejection in a regulatory T cell-independent manner, BALB/c mice inoculated with $0.5 \times 10^6$ CT26 subcutaneously in the flank were further randomised into mice treated with vehicle or with iTreg, PI-3065 (75 mg/kg, daily), after tumour reached between 50 and 100 mm$^3$. Maximal tumour burden permitted by CMU IACUC is 2000 mm$^3$.

**Coefficient of drug interaction (CI)**. It is calculated as follows: $CI = AB/(A \times B)$. AB is the ratio of the two-drug combination group to the control group. A or B is the ratio of the single-drug group to the control group. $CI < 1$ indicates synergism. $CI < 0.7$ indicates a significant synergistic effect. $CI = 1$ indicates additively, $CI > 1$ indicates antagonism[66].

**Statistical analysis**. Statistical analyses were performed using the Excel (ver. 2016) and GraphPad Prism software. Statistical significance in tumour growth is determined by two-way analysis of variance with Bonferroni posttest. Unless otherwise stated, all data are expressed as mean ± s.e.m.

**Reporting summary**. Further information on research design is available in the Nature Research Reporting Summary linked to this article.

## Data availability

The TCGA data used in this study are publicly available in cBioportal [https://www.cbioportal.org/]. All CCLE processed data sets are available at the CCLE portal [www.broadinstitute.org/ccle] and DepMap portal [http://www.depmap.org]. Whole-exome sequencing data and mRNA data of carcinogen-induced tumours from Lgr5+ intestinal stem cells in pp2a$^{-/-}$ mice are available with NIH/NCBI at SRA data set under accession number SRP162613 and at GEO data set under accession code GSE120241. RNA sequencing data of CT26-shppp2r1a and CT26-scr samples are available with NIH/NCBI at GEO data set under accession code GSE182571. The mass spectrometry data are available as Supplementary Data 1. TCGA data. TCR variable regions transcripts of CT26-shppp2r1a and CT26-scr samples are available with NIH/NCBI at GEO data set under accession code GSE189092. The remaining data are available within the Article, Supplementary Information, or Source Data file. Source data are provided with this paper.

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

## Acknowledgements

This work was supported by the "Drug Development Center, China Medical University" from The Featured Areas Research Center Program within the framework of the Higher Education Sprout Project by the Ministry of Education (MOE) in Taiwan. This work was financially supported by the Ministry of Science and Technology (MOST 106-2321-B-039 -003; 109-2321-B-039 -003), China Medical University (CMU110-Z-05), and China Medical University Hospital (DMR-108-BC-5). The funding sources had no involvement in study design, in the collection, analysis and interpretation of data, in the writing of the report, and in the decision to submit the article for publication.

## Author contributions

Y.-T.Y.: writing—original draft preparation, methodology, experiment conduction, data analysis; M.C.: experiment conduction, data analysis; P.-Y.W.: experiment conduction, data analysis; K.H., S.-F.C., K.C., W.C.: design and perform clinical studies, IRB protocol conduction, and tumour tissue microarray acquisition; S.-C.H.: writing—original draft preparation, writing—review and editing, conceptualisation, methodology, data analysis, funding acquisition.

## Competing interests

The authors declare no competing interests.
