## [Peer Review File · Nature Communications]

Reviewers' Comments:

Reviewer #1:

Remarks to the Author:

The present study by Yen et al. suggests inactivation of protein phosphatase 2A (PP2A) as a mechanism responsible for the induction of microsatellite instability (MSI). The authors suggest PP2A inactivation as a novel approach to enhance anti-tumor immune responses and to turn "cold" MSS tumors into "hot" MSI tumors.

The concept is interesting, but the manuscript suffers from several problems outlined below. Mainly, functional evidence for the hypothesis of MSI induction by PP2A inactivation is incomplete, and descriptive data from human and mouse tumors cannot support the claim of a causal link.

1. In the legend to Figure 5, the authors state that treatment with LB-100 was performed for 2 days. Length differences for BAT25 between LB-100-treated cells and untreated controls are extremely pronounced with more than 10 basepairs difference. Acknowledging previous studies and estimations for basepair deletion rates per cell division, this cannot be explained. Previous studies suggested mononucleotide peak patterns (precisely of the markers used in the present study) as molecular clocks. These previous studies consistently reported the emergence of MSI-indicating peaks only months after the onset of MMR deficiency, which is in stark contrast to the data presented here.

As this experiment is a crucial to convincingly demonstrate the functional connection between PP2A inactivation and MSI, thorough re-analysis is required. To assess this point, the authors should perform standardized time scale experiments, quantitatively examining changes of the peak patterns in parallel after knockout and LB-100 treatment.

2. Results of MSI analyses are presented poorly. Figure 5e: Shifts indicative of MSI seem to be present for some markers, most evidently BAT25. I have the following problems with this figure: (1) Lower right panels labeled "D2S123" do not correspond to D2S123 (dinucleotide marker), but show a mononucleotide allele pattern. (2) Peak patterns are cut, a larger range of allele sizes should be displayed to visualize all peaks (for example BAT40 for SW480 cannot be interpreted properly). (3) The authors should add arrows or asterisks to help with the interpretation of the MSI results in the corresponding figure legends. (4) Labeling of x and y axes is small and hard to read, scale units should be enlarged.

Similarly, Fig 2d does not indicate clear MSI in contrast to what is stated in the legend. This needs clarification. Also here, scale units are hard to read, but signal intensities appear low for GeneMapper data?

3. Abstract, Page 2, line 32: "however, the mechanism of MSI status development is unclear". This statement disregards the history of MSI cancer research and the fact that genetic and epigenetic alterations responsible for the MSI phenotype have been largely clarified; according to some recent studies, epigenetic silencing of MLH1, somatic biallelic MMR gene mutations, and the combination of first and second hit of the same MMR gene in Lynch syndrome together can explain the vast majority of MSI cancers. The authors only present correlation data suggesting a link between PP2A and MSI, no evidence is presented that PP2A inactivation is responsible for the natural occurrence of the MSI phenotype in human cancers. The increased proportion of MSI tumors among endometrial cancers harboring PP2A, SET, and CIP2A mutations may be coincidental or related to the fact that MMR deficiency is generally associated with high mutation burden (so that MMR deficiency is cause, not consequence). The conclusion that PP2A mutation status may help to predict ICB therapy response based on these data is not justified.

Minor points:

4. The authors use the cell lines SW-480 and SW-620 as examples for human CRC cell lines. Both cell lines are derived from the same patient and tumor, but derived at different time points. Therefore, one can expect that both lines will behave similarly regarding treatment response. What was the reasoning behind this selection?

5. Differences in gene expression as presented in Figure 2i are marginal for some candidates. Judging from the plots, data are not following a normal distribution (for example PPP2R1A in MSI), therefore non-parametrical tests should be used to assess significance.

6. Page 14, line 315: The authors claim that all MSI CRC belong to the consensus molecular subtype I. This is an overstatement, as in Guinney et al. 2015 only 75% of MSI CRC were classified as subtype I. Similarly, the MSS tumors are distributed over the subtypes, with only a smaller proportion showing features of subtype IV. The statement needs to be amended.

7. The following section (page 15, line 324) "Therefore, MSI CRC has the tumour microenvironment of type I CMS ..." contains confident statements about combination therapies, claiming for two scenarios that "there is no need to combine Treg inhibition". Those are theoretical considerations, reasonable or not, and should be indicated as such. In this context, studies demonstrating clinical benefit from combination of anti-PD-1 and anti-CTLA4 should be considered.

8. Page 7, line 162: Total HDAC2 levels and Rb levels are not demonstrated in Figure 3f.

9. Figure 3g: The difference in E2F1 levels are very small, however, dramatic differences in DNMT and MLH1 levels are observed. How can this be explained?

Reviewer #2:

Remarks to the Author:

The study shows an interesting connection of PP2A inactivation to MLH silencing and MSI and demonstrates that PP2A knockout promotes the therapeutic response to ICB. The study began with a mouse model showing depletion of ppp2r1a which encodes the PP2A A scaffold subunit increases immunogenicity, T cell infiltration and induction of MSI status. This piece of data in the mouse model is convincingly demonstrated to support the claim that PP2A indeed has a role to play in cancer immunity. The author went on to examine if the finding obtained from the mouse is clinically relevant. Using TCGA database, they claimed that the mRNA levels of PP2A endogenous inhibitors such as CIP2A and SET are significantly higher in CRC samples with MSI compared to that with MSS, while the PPP2R1A is lower. Protein analysis using CRC tissue array further shows the PPP2R1A is indeed lower in MSI tumors while the CIP2A and SET are higher. The subsequent experiment using the mouse model explored the mechanism. It showed that PPP2R1A co-immunoprecipitated the HDAC2 and Rb1 etc and PPP2R1a loss resulted in phosphorylation of HDAC2 and Rb1, leading to increased DNMT expression and MLH1 methylation and silencing. Finally, in a C26 syngenic CRC model, they showed that ppp2r1a knockdown increased response to anti-PD1 and a combination of a chemical inhibitor of PP2A with anti-PD1 also shows some better response in growth inhibition.

Major concerns

Although the mouse data documenting a role of PP2A (by PPP2R1a A structural unit) in regulating MLH1 expression is convincing, the clinical data supporting the clinical relevance of PP2A-PPP2R1A loss in relation to MSI status in CRC is questionable. First, as the endogenous inhibitor of PP2A, CIP2A and SET function to regulate PP2A activity, instead of PPP2R1A expression. The authors provided data showing that CIP2A and SET is weakly upregulated in MSI tumors compared to MSS tumors but this does not necessarily warrant a claim that PP2A activity is lower in MSI tumors. Showing PPP2R1a is lower in MSI tumors is strange as PPP2R1a expression in general is not downregulated in CRC. Instead, other B subunits of PP2A are widely silenced by DNA methylation in 90% of CRC. This suggests that the majority of the CRC have PP2A dysfunction as a tumor suppressor. Given that MSI tumor is only found in 10-15% of CRC, it is hard to believe that only the MSI tumors have the PP2A inactivation while most MSS tumors are not. The authors need to address this more carefully.

For the mechanistic study, the biochemical evidence showing HDAC2 and Rb1 are PP2A substrates are lacking. Although they are co-immunoprecipitated, it is necessary to demonstrate that they are the direct substrates of PP2A. PPP2R1a pulldown for an in vitro phosphatase assay using CT26 cell lysis as a substrate will substantiate the mechanistic claim.

The final data showing PP2A inhibitor sensitizing PD1 therapy is not impressive. I find it is hard to appreciate the claim of using PP2A inhibitor as a strategy to induce MSI and then sensitize ICB. As PP2A is widely inactivated in human CRC, it is not convincing that inhibition of PP2A can be a strategy for cancer therapy. Also, inhibiting PP2A as a tumor suppressor can cause lots of

oncogenic signaling, including RB E2F1 and many other known oncogenes. I am afraid that the therapeutic implication of using PP2A inhibitor in CRC has limited potential (though it has been reported before).

Also, the study focuses on colorectal cancer and it is not clear why the in vivo studies also included the triple-negative 4T1 and pancreatic model.

Reviewer #3:

Remarks to the Author:

The antigenicity and hence the immunogenicity of tumours is likely to be a major limiting factor in response to immunotherapy. Altering immunogenicity is a major challenge and this study is potentially important because it attempts to rise to this challenge by evaluating a pathway which may give rise to new antigens.

Whilst the experiments examining the impact of targeting PP2A on MSI status are compelling, the experiments examining effects on the immune response fall short of providing definitive answers. As such, the data as it stands is over-interpreted.

The histology shown in Figure 1 needs some improvement. It is difficult to understand why no CD4+ cells are observed when Foxp3+ cells are seen? What are the cells stained in the lower right panel?

In studies of human CRC (and other cancers) CD8+ T cells and Tregs are normally reported as positively correlating? This is because Tregs are induced when there is an immune response to suppress. This does not appear to be the case in the analysis shown here. Could the authors comment?

The tumour growth curves in Figure 4 show significant differences however these are only assessed for a short period of time (up to day 21). Did the study extend beyond this time-point? The comparison of lymphocyte numbers by IF staining must have been carried out on very small tumours where PP2A is absent. Smaller tumours often have more lymphocytes / g tumour compared to larger tumours thus the authors should provide more details and/or normalise for tumour size.

It is impossible to conclude that there are more neoantigens generated in these tumours without exome sequencing. This is the key missing piece of data. The number of TCRs alone is not sufficient evidence of neoantigen-driven clonal expansions.

The data with the small molecule inhibitor is less compelling. Also, despite the authors' claim that there is an effect beyond an impact of Tregs; this is not proven by the experiment carried out as it does not include the use of the PI-3065 plus anti-PD1 alone as a control. In addition, the authors should note that PI-3065 has effects beyond just direct effects on Tregs. It also affects effector T cells directly as well as monocytes. This experiment needs a re-think to include all necessary controls as well as a "cleaner" method of targeting Tregs.

Reviewers' comments:

Reviewer #1 (Remarks to the Author):

The present study by Yen et al. suggests inactivation of protein phosphatase 2A (PP2A) as a mechanism responsible for the induction of microsatellite instability (MSI). The authors suggest PP2A inactivation as a novel approach to enhance anti-tumor immune responses and to turn “cold” MSS tumors into “hot” MSI tumors.

The concept is interesting, but the manuscript suffers from several problems outlined below. Mainly, functional evidence for the hypothesis of MSI induction by PP2A inactivation is incomplete, and descriptive data from human and mouse tumors cannot support the claim of a causal link.

1. In the legend to Figure 5, the authors state that treatment with LB-100 was performed for 2 days. Length differences for BAT25 between LB-100-treated cells and untreated controls are extremely pronounced with more than 10 basepairs difference. Acknowledging previous studies and estimations for basepair deletion rates per cell division, this cannot be explained. Previous studies suggested mononucleotide peak patterns (precisely of the markers used in the present study) as molecular clocks. These previous studies consistently reported the emergence of MSI-indicating peaks only months after the onset of MMR deficiency, which is in stark contrast to the data presented here.

As this experiment is a crucial to convincingly demonstrate the functional connection between PP2A inactivation and MSI, thorough re-analysis is required. To assess this point, the authors should perform standardized time scale experiments, quantitatively examining changes of the peak patterns in parallel after knockout and LB-100 treatment.

Response: Thank you very much for your great comments. We have now provided more data to prove the causal link between PP2A loss with MSI induction and its relevance to human CRC. Western blotting analysis showed that treatment with different shRNAs against PPP2R1A or with different PP2A inhibitors (LB100 and LB102) for 2 or 7 days decreased MLH1 protein levels in SW620 and another human MSS CRC cell line, HT29 (Fig. 5d). Western blot analysis of SW620; HT29 transfected with indicated shRNAs, and treated with vehicle control or LB100/LB102. We have also now used commonly recognized method and definition to define whether the tumour was MSI¹. The panel of markers included D2S123, D5S346, D17S250 and BAT25. Compared with control cells, HT29 treated with PP2A inhibitors for 2 or 7 days or with different shRNAs against PPP2R1A showed changes in the length of all markers used (Fig. 5e and Fig. S11, S12). Moreover, the profiles of marker length changes caused by treatment with PP2A inhibitors (LB100) for 2 or 7 days or the use

of different shRNA treatments against PPP2R1A were almost similar. These data suggest that MSI induction caused by PP2A inhibition or knockdown occurs very rapidly and shares the similar profiles of MSI marker length changes. These findings were supported by MSI induction caused by hypoxia² and chemical agents³, when MSI induction happened 2 and 3 days after the treatment, respectively. Together, these data suggest the causal link between PP2A loss with MSI induction and its relevance to human CRC.

The reason that we did not use knockout to perform this experiment is that complete A α loss has no transformative properties. Additionally, in cancer patients, A α is found to be inactivated in a haploinsufficient manner. CRISPR/Cas9-mediated homozygous A α deletion resulted in decreased colony formation and tumour growth across multiple colorectal and endometrial cancer cell lines. This study further uncovered a mechanism by which PP2A A α regulates A β protein stability and activity and suggests why homozygous loss of A α is rarely seen in cancer patients²⁸.

2. Results of MSI analyses are presented poorly. Figure 5e: Shifts indicative of MSI seem to be present for some markers, most evidently BAT25. I have the following problems with this figure: (1) Lower right panels labeled "D2S123" do not correspond to D2S123 (dinucleotide marker), but show a mononucleotide allele pattern. (2) Peak patterns are cut, a larger range of allele sizes should be displayed to visualize all peaks (for example BAT40 for SW480 cannot be interpreted properly). (3) The authors should add arrows or asterisks to help with the interpretation of the MSI results in the corresponding figure legends. (4) Labeling of x and y axes is small and hard to read, scale units should be enlarged.

Response: Thank you very much for the great comments. Microsatellite instability analysis was performed using five National Cancer Institute consensus microsatellite markers (BAT25, BAT26, D2S123, D5S346 and D17S250)¹ as known as Bethesda panel. We have now provided a mini-table (Fig. 5e) and magnified original data (Fig. S11 and Fig. S12) that clearly indicate the changes in the size of the microsatellite markers of each treatment in the HT29 and SW620 human CRC. Especially, the predominant amplicon size bands (peaks) in mutant alleles are indicated as red number(s) and asterisk(s) in all sample traces. Shiftings comparing to control (CTR) are marked at right under each sample name.

Similarly, Fig 2d does not indicate clear MSI in contrast to what is stated in the legend. This needs clarification. Also here, scale units are hard to read, but signal intensities appear low for GeneMapper data?

Response: Thank you very much for the great comments. We have now provided clear MSI data in Fig. 2d. We divided the data of each marker into three panels and used different colors to clearly indicate the size change according to the colors used to mark the type of treatment. In addition, the scale unit has been clearly labelled on the tops of each marker.

3. Abstract, Page 2, line 32: “however, the mechanism of MSI status development is unclear”. This statement disregards the history of MSI cancer research and the fact that genetic and epigenetic alterations responsible for the MSI phenotype have been largely clarified; according to some recent studies, epigenetic silencing of MLH1, somatic biallelic MMR gene mutations, and the combination of first and second hit of the same MMR gene in Lynch syndrome together can explain the vast majority of MSI cancers. The authors only present correlation data suggesting a link between PP2A and MSI, no evidence is presented that PP2A inactivation is responsible for the natural occurrence of the MSI phenotype in human cancers. The increased proportion of MSI tumours among endometrial cancers harboring PP2A, SET, and CIP2A mutations may be coincidental or related to the fact that MMR deficiency is generally associated with high mutation burden (so that MMR deficiency is cause, not consequence).

Response: Thank you very much for the great comments.

1. We have revised the abstract as “Microsatellite-unstable (MSI) tumours are one of the few cancers that respond to immune checkpoint blockade (ICB) with genetic and epigenetic alterations well clarified; however, the mechanism of MSI status development is not well understood.” Please refer to Page 2, line 32 to 35 in the revised version.
2. Protein phosphatase 2A (PP2A) is a tumour suppressor that regulates many signaling pathways²⁹⁻³¹, and its loss of function has been associated with cell transformation³². PP2A has been directly implicated in the negative regulation of double-strand break DNA repair proteins³³. Consistent with the idea those protein phosphatases are not just negative regulators of DNA repair signaling, selective inhibition of PP2A activity impairs DNA repair^{34,35}. PP2A has been suggested or confirmed to dephosphorylate over 300 substrates including MLH1, PMS1, and PMS2³⁶. We also use experiments to prove that pp2a inhibitor can reduce MLH1 expression and MSI status (Fig. 5e, Fig. S11 and Fig. S12).
3. We further demonstrated the positive correlation between mutation count and CIP2A and SET mRNA levels (Unshown Figure. 1). The correlation coefficients were $r = 0.13$ ($P = 0.003$) and $r = 0.18$ ($P < 0.00005$) for CIP2A and SET, respectively.

Unshown Figure. 1. The mRNA levels of CIP2A and SET mRNA level (Y-axis) positively correlated with total mutation count (mean) (X-axis) for TCGA-colorectal cancer samples.

We used the classification and regression trees (CART), a powerful approach optimizing the cutoff point of independent variables for predicting dependent variable used in medical data sets³⁷, to divide the CIP2A or SET data into high-level and low-level subgroups. The cut-off values of CIP2A and SET data were calculated as 440.953 and 11716.08, respectively. The percentages of the subgroups with high and low levels of CIP2A were 16.22% and 83.78%, respectively. In addition, the mutation count of the subgroup with high CIP2A was significantly higher than that of the subgroup with low CIP2A ($P = 0.015$) (Unshown Table 1). Similarly, the percentages of the subgroups with high and low levels of SET were 3.42% and 96.58%, respectively. Moreover, the mutation count in the high SET subgroup was higher than that in the low SET subgroup, although not significant. (Unshown Table 1). In the MSK-IMPACT cohort, including the clinical and genomic data of 1,661 advanced cancer patients treated with ICB³⁸, tumours with *PPP2R1A* mutation accounted for 1.4%, which was associated with increased tumour mutation burden score and mutation count and better overall survival status (Fig. S15). Together, these data indicate that PP2A is not widely inactivated in colorectal tumours, and inhibition of PP2A may be a strategy for colorectal cancer treatment.

Unshown Table 1. Mutation count between the subgroups for CIP2A and SET

CIP2A	Cut point	N (%)	Mutation count (Mean, SE)	P value
		≥ 440.953	85 (16.22)	834, 207

	< 440.953	439 (83.78)	313, 37	
SET	Cut point	N (%)	Mutation count (Mean, SE)	P value
	≥ 11716.08	18 (3.42)	1262, 580	0.1415
	< 11716.08	508 (96.58)	366, 43	

The conclusion that PP2A mutation status may help to predict ICB therapy response based on these data is not justified.

Response: Thank you very much for the great comments.

A previous study analyzed the clinical and genomic data of 1,661 advanced cancer patients treated with ICB, whose tumours underwent targeted next-generation sequencing (MSK-IMPACT)³⁸, and showed that higher somatic tumour mutational burden (highest 20% in each histology) can predict survival after immunotherapy across multiple cancer types. Given its clinical and practical importance, we have undertaken a reanalysis of these data and showed that PPP2R1A mutation (1.4%) was associated with an increased tumour mutation burden score and mutation count, and a better overall survival status (Fig. S15a, b). Moreover, the median survival time and the univariate Cox regression hazard ratio of patients with PPP2R1A-mutated tumours were much better than those of patients with PPP2R1A-non-mutated tumours (Fig. S15c). The pan-cancer nature of this biomarker probably reflects the fundamental mechanisms by which ICB functions. These data also support the hypothesis that PPP2R1A, SET, and CIP2A mutations or altered mRNA levels are associated with higher mutation burden and MSI status and help predict response to ICB.

Minor points:

4. The authors use the cell lines SW-480 and SW-620 as examples for human CRC cell lines. Both cell lines are derived from the same patient and tumor, but derived at different time points. Therefore, one can expect that both lines will behave similarly regarding treatment response. What was the reasoning behind this selection?

Response: Thank you very much for your great comments. We apologize for this mistake and further explain the reason for using these two cell lines. Although SW480 (primary) and SW620 (lymph node) cell lines derived from primary tumour and metastasis from the same patient and carried identical mutation profiles, but had epigenetic differences³⁹. According to your comments, we have also now repeated the use of PP2A inhibitors, not only LB-100 but also LB-102, to induce MSI in another MSS human CRC cell line, HT29. As shown in Fig. 5e, Fig. S11 and S12, treatment with LB100 or LB-102 induced MSI both at 2 and 7 days.

5. Differences in gene expression as presented in Figure 2i are marginal for some candidates. Judging from the plots, data are not following a normal distribution (for example PPP2R1A in MSI), therefore non-parametrical tests should be used to assess significance.

Response: Thank you very much for the great comments. A normality test⁴⁰ was used to demonstrate that the data regarding CIP2A and SET are well-modeled by a normal distribution and the parametrical test (unpaired t-test) was used to examine significant values. The p values for CIP2A and SET were p=0.0096 and p=0.05, respectively. Because PPP2R1A data were not normally distributed, we therefore used the non-parametrical test (Mann Whitney U test: The procedure PROC NPAR1WAY in SAS version 9.4) to reanalyze the difference in PPP2R1A values between MSI and MSS tumours. However, the difference is not significant (p=0.1576.) The statistical trend is similar to the results shown in Fig. 2j, where the p values for CIP2A and SET were found significant, while the p value for PPP2R1A was not found significant.

6. Page 14, line 315: The authors claim that all MSI CRC belong to the consensus molecular subtype I. This is an overstatement, as in Guinney et al. 2015 only 75% of MSI CRC were classified as subtype I. Similarly, the MSS tumors are distributed over the subtypes, with only a smaller proportion showing features of subtype IV. The statement needs to be amended.

Response: Thank you very much for the great comments. We have now modified this part as “Based on the consensus molecular subtypes (CMSs) of colorectal cancer⁴¹, majority (~75%) of MSI tumours belong to type I CMS (characterized by increased expression of genes associated with a diffuse immune infiltrate, mainly composed of TH1 and cytotoxic T cells), while tumours caused by *Apc*^{fl/fl}, *Kras*^{LSL-G12D}, *Tgfbr2*^{fl/fl} and *Trp53*^{fl/fl}, belong to type IV CMS (characterized by TGFβ-activated stroma)⁴² are corresponding to MSS, suggesting the tumour itself is the dominant force shaping the tumours microenvironment. ” Please refer to Page 15 in line 339-344 in the revised version.

7. The following section (page 15, line 324) “Therefore, MSI CRC has the tumour microenvironment of type I CMS ...” contains confident statements about combination therapies, claiming for two scenarios that “there is no need to combine Treg inhibition”. Those are theoretical considerations, reasonable or not, and should be indicate as such. In this context, studies demonstrating clinical benefit from combination of anti-PD-1 (LB-100?) and anti-CTLA4 should be considered.

Response: Thank you very much for the great comments. A previous study analyzed the clinical and genomic data of 1,661 advanced cancer patients treated with ICB (including **CTLA-4**, PD-1/PDL-1 and Combo group), whose tumours underwent targeted next-generation sequencing (MSK-IMPACT)³⁸, and showed that higher somatic tumour mutational burden (highest 20% in each histology) can predict survival after immunotherapy across multiple cancer types. Given its clinical and practical importance, we have undertaken a reanalysis of these data and showed that PPP2R1A mutation (1.4%) was associated with an increased tumour mutation burden score and mutation count, and a better overall survival status (Fig. S15a, b). Moreover, the median survival time and the univariate Cox regression hazard ratio of patients with PPP2R1A-mutated tumours were much better than those of patients with PPP2R1A-non-mutated tumours (Fig. S15c). The pan-cancer nature of this biomarker probably reflects the fundamental mechanisms by which ICB functions. These data also support the hypothesis that PPP2R1A, SET, and CIP2A mutations or altered mRNA levels are associated with higher mutation burden and MSI status and help predict response to ICB. In response to your question about the combination of LB-100 and anti-CTLA4, we believe that the combination of LB-100 and anti-CTLA may improve the therapeutic effect.

8. Page 7, line 162: Total HDAC2 levels and Rb levels are not demonstrated in Figure 3f.

Response: Thank you very much for the great comments. We apologize for this mistake. We have now added total HDAC2 and Rb level in Fig. 3f.

9. Figure 3g: The difference in E2F1 levels are very small, however, dramatic differences in DNMT and MLH1 levels are observed. How can this be explained?

Response: Thank you very much for the great comments. We double checked the data and have now provided clear E2F1 data in Fig. 3g. The new data are more in line with the levels of DNMT 3A/3B and MLH1 in the revised version.

Reviewer #2 (Remarks to the Author):

The study shows an interesting connection of PP2A inactivation to MLH silencing and MSI and demonstrates that PP2A knockout promotes the therapeutic response to ICB. The study began with a mouse model showing depletion of ppp2r1a which encodes the PP2A A scaffold subunit increases immunogenicity, T cell infiltration and induction of MSI status. **This piece of data in the mouse model is convincingly demonstrated to support the claim that PP2A indeed has a role to play in cancer immunity.** The author went on to examine if the finding obtained from the mouse is clinically relevant. Using TCGA database, they claimed that the mRNA levels of PP2A

endogenous inhibitors such as CIP2A and SET are significantly higher in CRC samples with MSI compared to that with MSS, while the PPP2R1A is lower. Protein analysis using CRC tissue array further shows the PPP2R1A is indeed lower in MSI tumors while the CIP2A and SET are higher. The subsequent experiment using the mouse model explored the mechanism. It showed that PPP2R1A co-immunoprecipitated the HDAC2 and Rb1 etc and PPP2R1a loss resulted in phosphorylation of HDAC2 and Rb1, leading to increased DNMT expression and MLH1 methylation and silencing. Finally, in a C26 syngenic CRC model, they showed that ppp2R1a knockdown increased response to anti-PD1 and a combination of a chemical inhibitor of PP2A with anti-PD1 also shows some better response in growth inhibition.

Response: Thank you very much for your wonderful summary of our research and the relevant favorable comments about the mouse model data.

Major concerns

Although the mouse data documenting **a role of PP2A (by PPP2R1a A structural unit) in regulating MLH1 expression is convincing**, the clinical data supporting the clinical relevance of PP2A-PPP2R1A loss in relation to MSI status in CRC is questionable. First, as the endogenous inhibitor of PP2A, CIP2A and SET function to regulate PP2A activity, instead of PPP2R1A expression. The authors provided data showing that CIP2A and SET is weakly upregulated in MSI tumors compared to MSS tumors but this does not necessarily warrant a claim that PP2A activity is lower in MSI tumors. Showing PPP2R1a is lower in MSI tumors is strange as PPP2R1a expression in general is not downregulated in CRC. Instead, other B subunits of PP2A are widely silenced by DNA methylation in 90% of CRC. This suggests that the majority of the CRC have PP2A dysfunction as a tumor suppressor. Given that MSI tumor is only found in 10-15% of CRC, it is hard to believe that only the MSI tumors have the PP2A inactivation, while most MSS tumors are not. The authors need to address this more carefully.

Response: Thank you very much for your great comments and kindly mentioned that “Instead, other B subunits of PP2A are widely silenced by DNA methylation in 90% of CRC.” The reference you referred we think may be the one published in Cancer Cell, 18:459-471, 2010. “B55 β -Associated PP2A Complex Controls PDK1-Directed Myc Signaling and Modulates Rapamycin Sensitivity in Colorectal Cancer”.

(1) This study “found that PPP2R2B, encoding B55 β , is the only subunit that is consistently downregulated or silenced in all examined CRC cell lines, but not in the normal colon mucosa samples”. Specifically, Figure 1B in the Cancer Cell paper shows that B55 β (PPP2R2B) is one of the few genes upregulated in primary tumour compared with normal tissue. However, PPP2R1A A subunit, PPP2CA C

subunit, and other B subunits, such as *PPP2R2A* (*B55α*), *PPP2R1B* (*PR65β*), *PPP2R3B* (*PR70*), *PPP2R5C* (*B56γ*), *PPP2R5D* (*B56δ*), are obviously upregulated in primary tumour compared with normal tissue⁴. These data show that compared with normal tissues, the main A and C subunits and most of the B subunits of PP2A are up-regulated in primary tumours. Because RNAi against Aα of PP2A decreased total PP2A activity^{5,6}, these data may imply an increase in PP2A activity in tumours⁴. However, this comparison was made between the primary tumour and each corresponding normal tissue, and has nothing to do with individual tumour differences. For example, SET was found overexpressed in 13 out the 21 tumour samples compared with corresponding normal tissue⁷, however, SET overexpression was only detected in 15.4% of 247 CRC patients without metastatic disease at diagnosis⁸. Need to be noted, the observed PP2A B subunit (*PPP2R2B*) silencing in the Cancer Cell paper was a result of comparison between the primary tumour and each corresponding normal tissue⁴.

- (2) Instead, we used TCGA data to examine the correlations of different immune cell infiltration with *PPP2R1A* and endogenous PP2A inhibitors in different individual tumours. We found that the expression of endogenous PP2A inhibitors, *CIP2A*, and *SET* correlated positively with the infiltration of CD8+ T cells and CD20+ B cells and negatively with FOXP3+ Treg cells (Fig. 1c). Contrastingly, the expression of *PPP2R1A* correlated negatively with CD8+ T cells but positively with FOXP3+ Treg cells (Fig. 1c). Similar findings were observed in human rectal cancers (TCGA) (Fig. 1d).
- (3) Moreover, long-term culture may induce changes in DNA methylation⁹, even though the cells studied here are human mesenchymal stem cells. Because *PPP2R2B* hypermethylation causes acquired apoptosis deficiency in activated T cells of systemic autoimmune diseases¹⁰, it is possible to induce cloned cells after long-term culture. Actually, this argument is supported by the data shown in the Cancer Cell paper⁴, where obvious *PPP2R2B* hypermethylation has been observed in CRC cell lines (Fig. 1E: *HCT116*, *RKO*....that only exhibited methylated band) compared to the primary CRC tumour samples (Fig. 1F: *2T*, *4T*, *12T*...that exhibited both unmethylated and methylated bands). Therefore, the increase in *PPP2R2B* DNA methylation levels of CRC cell lines compared with the primary tumour cells can easily be explained as the contribution of long-term culture. As expected, frequent aberrant DNA methylation of *PPP2R2B* was observed in primary tumour tissues of ductal carcinoma in situ and early invasive breast cancer. *PPP2R2B* DNA has been shown to be hypomethylated in cancers with TP53 mutations¹¹. These data suggest long-term culture induces changes in DNA methylation, which may help select cells with acquired apoptosis resistance

through PPP2R2B hypermethylation.

- (4) The survival and proliferation of established CRC cells (SW620 and HT29 lines) and primary human colon cancer cells can be suppressed by LB-100 that inhibited PP2A activity and activated AMPK signaling both *in vitro* and *in vivo*¹². In addition, the self-renewal and sphere formation HT29 cell line and primary human colon cancer cells can be suppressed by silibinin¹³ that inhibited the PP2Ac/AKT Ser473/mTOR pathway. Similarly, we used PP2A inhibitors, LB100 and LB102 to suppress PP2A activity (Fig. S10) and thereby induced MSI status (Fig. 5e and Fig. S11, S12). Together, these data indicate that PP2A activity and its downstream pathways of CRC cell lines and primary human CRC cells can still be manipulated by agents that inhibit PP2A activity.

For the mechanistic study, the biochemical evidence showing HDAC2 and Rb1 are PP2A substrates are lacking. Although they are co-immunoprecipitated, it is necessary to demonstrate that they are the direct substrates of PP2A. PPP2R1a pulldown for an *in vitro* phosphatase assay using CT26 cell lysis as a substrate will substantiate the mechanistic claim.

Response: Thank you very much for your great comments. We have now demonstrated that HDAC2 and Rb1 are the direct substrates of PP2A. CT26 cells were treated without (CTR) and with LB100, small-molecule inhibitor of PP2A, followed by PPP2R1a pulldown of the cell lysates for an *in vitro* phosphatase assay and western blotting. The data showed that the PPP2R1A pulldown in LB100-treated cell lysate exhibited decreased PP2A activity (Fig. S5a) and increased pRb1 and pHDAC2 levels (Fig. S5b). The biochemical evidence showing that HDAC2 and Rb are direct PP2A substrates. We have also revised manuscript as "To further demonstrate that Rb and HDAC2 were the direct substrates of PP2A, CT26 cells were treated without (CTR) and with LB100, a small molecule inhibitor of PP2A¹⁴, followed by Ppp2r1a pulldown of the cell lysates for an *in vitro* phosphatase assay and western blotting. The data showed that the Ppp2r1a pulldown in LB100-treated cell lysate exhibited decreased PP2A activity (Fig. S5a) and increased pRb and pHDAC2 levels (Fig. S5b). The biochemical evidence showing that HDAC2 and Rb1 are direct PP2A substrates." Please refer to Page 7 in line 159-164 in the revised version.

The final data showing PP2A inhibitor sensitizing PD1 therapy is not impressive. I find it is hard to appreciate the claim of using PP2A inhibitor as a strategy to induce MSI and then sensitize ICB. As PP2A is widely inactivated in human CRC, it is not convincing that inhibition of PP2A can be a strategy for cancer therapy.

Response: Thank you very much for your great comment. We have used TCGA

colorectal cancer samples (n=594) to demonstrate that compared with MSS cancers, MSI cancers have increased CIP2A and SET mRNA levels, but reduced PPP2R1A mRNA levels (Fig. 2i). We further demonstrated the positive correlation between mutation count and CIP2A and SET mRNA levels (Unshown Figure. 1). The correlation coefficients were $r = 0.13$ ($P = 0.003$) and $r = 0.18$ ($P < 0.00005$) for CIP2A and SET, respectively.

Unshown Figure. 1. The mRNA levels of CIP2A and SET mRNA level (Y-axis) positively correlated with total mutation count (mean) (X-axis) for TCGA-colorectal cancer samples.

We used the classification and regression trees (CART), a powerful approach optimizing the cutoff point of independent variables for predicting dependent variable used in medical data sets³⁷, to divide the CIP2A or SET data into high-level and low-level subgroups. The cut-off values of CIP2A and SET data were calculated as 440.953 and 11716.08, respectively. The percentages of the subgroups with high and low levels of CIP2A were 16.22% and 83.78%, respectively. In addition, the mutation count of the subgroup with high CIP2A was significantly higher than that of the subgroup with low CIP2A ($P = 0.015$) (Unshown Table 1). Similarly, the percentages of the subgroups with high and low levels of SET were 3.42% and 96.58%, respectively. Moreover, the mutation count in the high SET subgroup was higher than that in the low SET subgroup, although not significant. (Unshown Table 1). In the MSK-IMPACT cohort, including the clinical and genomic data of 1,661 advanced cancer patients treated with ICB³⁸, tumours with *PPP2R1A* mutation accounted for 1.4%, which was associated with increased tumour mutation burden score and mutation count and better overall survival status (Fig. S15). Together, these data indicate that PP2A is not

widely inactivated in colorectal tumours, and inhibition of PP2A may be a strategy for colorectal cancer treatment.

Unshown Table 1. Mutation count between the subgroups for CIP2A and SET

CIP2A	Cut point	N (%)	Mutation count (Mean, SE)	P value
	≥ 440.953	85 (16.22)	834, 207	0.0154
	< 440.953	439 (83.78)	313, 37	
SET	Cut point	N (%)	Mutation count (Mean, SE)	P value
	≥ 11716.08	18 (3.42)	1262, 580	0.1415
	< 11716.08	508 (96.58)	366, 43	

Also, inhibiting PP2A as a tumor suppressor can cause lots of oncogenic signaling, including RB E2F1 and many other known oncogenes. I am afraid that the therapeutic implication of using PP2A inhibitor in CRC has limited potential (though it has been reported before).

Response: Thank you very much for your great comment. Although the safety of LB100 has partly addressed in the phase I-II clinical trial¹⁴, there is a concern that inhibiting PP2A, a tumour suppressor, can cause lots of oncogenic signaling. We have addressed the limitations of current study in the discussion section and proposed some tumour-targeting and controlled-release drug carrier systems, such as liposome or nanocages, can solve these limitations. We have now further addressed this issue in the discussion section. “To be noted, inhibiting PP2A, a tumour suppressor, can cause lots of oncogenic signals in normal tissues, thereby limiting the therapeutic potential of PP2A inhibition in cancer treatment. Similarly, this problem can be solved by controlled delivery of therapeutic agents to tumours.” Please refer to Page 16 in line 368-370.

Also, the study focuses on colorectal cancer and it is not clear why the in vivo studies also included the triple-negative 4T1 and pancreatic model.

Response: Thank you very much for your great comment. Both of triple-negative breast cancer³² and pancreatic cancer⁴⁹ have low incidence of dMMR/MSI-H tumours and poorly respond to monotherapy with antibodies against PD1 or PD-L1^{50,51}. However, immune checkpoint blockade is a FDA-approved tissue-agnostic drug for the treatment of MSI-high solid tumours⁵². Pancreatic cancer was also included in 12 tumour types in the cohort of 86 MSI-high tumour patients and responded to anti-PD1 monotherapy⁵³. We therefore expanded the application of current results to the treatment of triple-negative 4T1 and pancreatic model. Our results show that the combined use of LB100 in 4T1 and Pan-18 tumour models increased the

sensitivity of ICB by increasing the mutation burden and inducing MSI status (Fig. 5a and Fig. S8).

Reviewer #3 (Remarks to the Author):

The antigenicity and hence the immunogenicity of tumours is likely to be a major limiting factor in response to immunotherapy. Altering immunogenicity is a major challenge and **this study is potentially important because it attempts to rise to this challenge by evaluating a pathway which may give rise to new antigens.**

Whilst the experiments examining the impact of targeting PP2A on MSI status are compelling, the experiments examining effects on the immune response fall short of providing definitive answers. As such, the data as it stands is over-interpreted.

Response: Thank you very much for your great comments. We have answered the following questions, which will help solve the problems you raised here.

The histology shown in Figure 1 needs some improvement. It is difficult to understand why no CD4+ cells are observed when Foxp3+ cells are seen? What are the cells stained in the lower right panel?

Response: Thank you very much for your great comments. We apologize for this mistake. The original figure for Foxp3 was misplaced with picture taken from distal small intestine that was filled with Treg attracted by CCR5¹⁵. The other figures in Fig. 1a were from colon (large intestine). In the revised Fig. 1a, all pictures were taken from colon. The new figures show that very few CD4+ or Foxp3+ cells were observed in the histology of control tissues (Fig. 1a).

The cells stained in the lower right panel are CD20+ B cell aggregates that were only observed in the tissue of ppp2r1a loss tumours. Tumour-infiltrating B cells have recently been identified as cellular components of tertiary lymphoid structures (TLSs) in the tumour, which are associated with better respond to immunotherapy¹⁹. The CD20+ B cell aggregates have been identified in melanoma and sarcoma (three papers side by side published in Nature), which are associated with an increased chance that patients' tumours would respond to immunotherapy¹⁶⁻¹⁸.

We further chose some figures published in the Nature papers and merged into the following figure for your reference.

Representative figures of B cell aggregates in TLS ,chosen from papers published by (a) Cabrita¹⁶ *et al.*, (b) Helmink¹⁸ *et al.* and (c) Petitprez¹⁷ *et al.* All demonstrate the presence of B cell aggregates contained in the TLS of human (a, b) melanoma and (c) sarcoma tissues.

In studies of human CRC (and other cancers) CD8+ T cells and Tregs are normally reported as positively correlating? This is because Tregs are induced when there is an immune response to suppress. This does not appear to be the case in the analysis shown here. Could the authors comment?

Response: Thank you very much for the great comments. We used TCGA colorectal cancer data that mainly include the transcriptome of the tumour per se. In other word, we analyzed the intratumoural CD8+ T cells and Tregs. In fact, the ratio of intratumoural CD8+ T/FOXP3+ cells in different colorectal tumours is very different, which is a predictive marker for the survival of colorectal cancer patients⁵⁴⁻⁵⁶. Intraepithelial lymphocytes were defined as lymphocytes located within tumour cell nests that may be the area used for transcriptome analysis. It has been noted that CD8 expression is detected in the epithelium in 100% of the cases, while FOXP3 expression is not or only sporadically present in the tumour epithelium⁵⁵. Therefore, the lack of a positive correlation between CD8+ T cells and Tregs in our data may be due to the specimens used for analysis.

The tumour growth curves in Figure 4 show significant differences however these are only assessed for a short period of time (up to day 21). Did the study extend beyond this time-point? The comparison of lymphocyte numbers by IF staining must have been carried out on very small tumours where PP2A is absent. Smaller tumours often have more lymphocytes / g tumour compared to larger tumours thus the authors should provide more details and/or normalise for tumour size. Response: Thank you very much for the great comments. Accordingly, we have now extended the study from 21 days to 28 and 35 days. Extension of the study does not change the results. Ppp2r1a knockdown inhibited tumour growth in the presence of anti-PD1, compared

to cells transfected with control shRNAs (WT CT26) (Fig. 4c). Although WT CT26 did not respond to anti-PD1 treatment (Fig. 4d), CT26 with Ppp2r1a knockdown responded to anti-PD1 treatment (Fig. 4e), suggesting Ppp2r1a knockdown sensitises CT26 to anti-PD1 treatment (Fig. 4e). Moreover, increased levels of CD8+ tumour-infiltrating T cells were found in tumours formed by CT26 cells with Ppp2r1a knockdown compared to those formed by WT CT26 cells (Fig. 4f).

We further provided the detailed data of CD8+ tumour-infiltrating T cells in tumours formed by CT26 with Ppp2r1a knockdown at different time-points from 21 to 35 days, when the tumours were found to become larger. After immunofluorescence staining, we processed the images to analyse the numbers of positive signals using TissueQuest software (TissueGnostics)²⁰. In order to normalise for tumour size, we also divided the numbers of CD8+ tumour-infiltrating T cells by the tumour weights (Fig. 4f Right)^{21,22}. However, these data did not show that smaller tumours had more lymphocytes/ g tumour compared to larger tumours.

It is impossible to conclude that there are more neoantigens generated in these tumours without exome sequencing. This is the key missing piece of data. The number of TCRs alone is not sufficient evidence of neoantigen-driven clonal expansions.

Response: Thank you very much for the great comments. Although there is a plethora of neoantigen discovery pipelines based on genetic information to predict epitopes, the current pipelines are human-centered and are therefore mainly designed for clinical use. Recently, NAP-CNB²⁷, a novel bioinformatic pipeline, has been developed to directly estimate H-2 peptide ligands from murine tumour samples, and its area under the curve (AUC) is equal to or better than the state-of-the-art methods. Moreover, this pipeline also has a neural network model of the binding affinity prediction function. We therefore used the NAP-CNB²⁷ pipeline to identify potential tumour neoantigens. The detailed method has been added in the "Materials and methods" section. We have also modified the manuscript as below: "To demonstrate that Ppp2r1a knockdown converted cold tumours into hot tumours by increasing neoantigen, we submitted the RNA-seq data of CT26-shppp2r1a and CT26-scr tumour samples, integrated in the fastq.gz files, and applied the NAP-CNB²⁷ to predict neoantigens. A total of 270 missense transcripts, corresponding to 220 genes, shared by three CT26-shppp2r1a tumours but not found in the CT26-scr tumour were identified (Fig. 4g). The software also generated a ranking of putative neoantigens that are common in the three CT26-shppp2r1a tumours samples. The 30 top-scoring putative neoepitopes are shown in Table S5." Please refer to Page 9 in line 193-200 in the revised version.

The data with the small molecule inhibitor is less compelling. Also, despite the authors' claim that there is an effect beyond an impact of Tregs; this is not proven by the experiment carried out as it does not include the use of the PI-3065 plus anti-PD1 alone as a control. In addition, the authors should note that PI-3065 has effects beyond just direct effects on Tregs. It also affects effector T cells directly as well as monocytes. This experiment needs a re-think to include all necessary controls as well as a "cleaner" method of targeting Tregs.

Response: Thank you very much for the great comments. We have now added the use of PI-3065 plus anti-PD1 alone as a control. The data showed that there is no significant difference between PI-3065 and PI-3065 plus anti-PD1 (Fig. 5b).

We have also strengthened the rationales of using PI-3065 to block mouse Treg-mediated immunosuppression.

- (1) The reason we did not use Foxp3- mutant scurfy mice or Foxp3-null mice or Treg-deficient mice for this study is that these mice suffered from the lethal lymphoproliferative autoimmune syndrome and become moribund at approximately 4 weeks of age²³. Although depletion of Treg cells by neonatal thymectomy, adoptive transfer of naive T cell samples depleted of Treg cells into lymphopenic hosts or treatment of mice with antibodies specific for CD25, results in a much milder and more slowly progressing disease²⁴, however, Treg cells are also critical in self-tolerance and prevent catastrophic autoimmunity throughout the lifespan of mice. We therefore gave up using the methods mentioned above.
- (2) Instead, we chose to use p110 δ inactivation that has been successfully demonstrated to block Treg-mediated immune suppression in mice carrying solid tumours²⁵. Notably, long-term administration of PI-3065, a small molecule inhibitor with selectivity for p110 δ , to mice was well tolerated and did not induce weight loss²⁵.
- (3) There are concerns that the inactivation of p110 δ in Treg cells will indirectly release CD8 cytotoxic T cells and induce tumour regression, and the inactivation of p110 δ will also block the intrinsic immunosuppression of PMN-MDSCs (Ly6G^{high}), leading to reduced tumour growth. However, it has also been reported that inhibiting p110 δ in cancer might impair cytotoxic T cells and negatively impact on cancer immune surveillance²⁶. Previous data²⁵ show that although p110 δ blockade reduces the effectiveness of cytotoxic T cells, it also overrides Treg- and probably also MDSC-mediated suppression of anti-tumour immune responses, enabling even weakened CTLs to successfully attack tumours. Thus, p110 δ is apparently more essential for Treg rather than effector T-cell responses

against cancer cells.

- (4) To show specific blocking of Treg by using the p110 δ inhibitor PI-3065, we first demonstrated that p110 δ was only expressed in Treg (Foxp3+), but not expressed in CD8+ or PMN-MDSCs (Ly6G^{high}) in the CT26 tumour microenvironment. We then used PI-3065 to block regulatory T cell-mediated immune suppression in mice²⁵, and thereby we could study the effect of anti-PD1 plus PP2A inhibition on tumour killing without Treg interference. “To prove that LB100 sensitised tumour cells to ICB therapies regardless of its Treg inhibitory activity, we first showed the expression of p110 δ in Treg (Foxp3+), but not in CD8+ or polymorphonuclear myeloid-derived suppressor cell (PMN-MDSC) (Ly6G^{high}) in the CT26 tumour microenvironment (Fig. S9). We then used the p110 δ inhibitor PI-3065 to block Treg-mediated immune suppression in mice²⁵, and showed that the therapeutic effects of the combination of LB100 and anti-PD1 on reducing tumour growth and enhancing survival were also observed in the presence of PI-3065 (Fig. 5b).” Please refer to Page 10 in line 222-229 in the revised version.

References:

- 1 Boland, C. R. *et al.* A National Cancer Institute Workshop on Microsatellite Instability for cancer detection and familial predisposition: development of international criteria for the determination of microsatellite instability in colorectal cancer. *Cancer Res* **58**, 5248-5257 (1998).
- 2 Rodriguez-Jimenez, F. J., Moreno-Manzano, V., Lucas-Dominguez, R. & Sanchez-Puelles, J. M. Hypoxia causes downregulation of mismatch repair system and genomic instability in stem cells. *Stem Cells* **26**, 2052-2062, doi:10.1634/stemcells.2007-1016 (2008).
- 3 Saletta, F. *et al.* Exposure to the tobacco smoke constituent 4-aminobiphenyl induces chromosomal instability in human cancer cells. *Cancer Res* **67**, 7088-7094, doi:10.1158/0008-5472.CAN-06-4420 (2007).
- 4 Tan, J. *et al.* B55beta-associated PP2A complex controls PDK1-directed myc signaling and modulates rapamycin sensitivity in colorectal cancer. *Cancer Cell* **18**, 459-471, doi:10.1016/j.ccr.2010.10.021 (2010).
- 5 Strack, S., Cribbs, J. T. & Gomez, L. Critical role for protein phosphatase 2A heterotrimers in mammalian cell survival. *J Biol Chem* **279**, 47732-47739, doi:10.1074/jbc.M408015200 (2004).
- 6 Flegg, C. P. *et al.* Nuclear export and centrosome targeting of the protein phosphatase 2A subunit B56alpha: role of B56alpha in nuclear export of the catalytic subunit. *J Biol Chem* **285**, 18144-18154, doi:10.1074/jbc.M109.093294 (2010).

- 7 Cristobal, I. *et al.* PP2A inhibition is a common event in colorectal cancer and its restoration using FTY720 shows promising therapeutic potential. *Mol Cancer Ther* **13**, 938-947, doi:10.1158/1535-7163.MCT-13-0150 (2014).
- 8 Cristobal, I. *et al.* Deregulation of SET is Associated with Tumor Progression and Predicts Adverse Outcome in Patients with Early-Stage Colorectal Cancer. *J Clin Med* **8**, doi:10.3390/jcm8030346 (2019).
- 9 Bork, S. *et al.* DNA methylation pattern changes upon long-term culture and aging of human mesenchymal stromal cells. *Aging Cell* **9**, 54-63, doi:10.1111/j.1474-9726.2009.00535.x (2010).
- 10 Madera-Salcedo, I. K. *et al.* PPP2R2B hypermethylation causes acquired apoptosis deficiency in systemic autoimmune diseases. *JCI Insight* **5**, doi:10.1172/jci.insight.126457 (2019).
- 11 Muggenrud, A. A. *et al.* Frequent aberrant DNA methylation of ABCB1, FOXC1, PPP2R2B and PTEN in ductal carcinoma in situ and early invasive breast cancer. *Breast Cancer Res* **12**, R3, doi:10.1186/bcr2466 (2010).
- 12 Dai, C. *et al.* Targeting PP2A activates AMPK signaling to inhibit colorectal cancer cells. *Oncotarget* **8**, 95810-95823, doi:10.18632/oncotarget.21336 (2017).
- 13 Wang, J. Y., Chang, C. C., Chiang, C. C., Chen, W. M. & Hung, S. C. Silibinin suppresses the maintenance of colorectal cancer stem-like cells by inhibiting PP2A/AKT/mTOR pathways. *J Cell Biochem* **113**, 1733-1743, doi:10.1002/jcb.24043 (2012).
- 14 Chung, V. *et al.* Safety, Tolerability, and Preliminary Activity of LB-100, an Inhibitor of Protein Phosphatase 2A, in Patients with Relapsed Solid Tumors: An Open-Label, Dose Escalation, First-in-Human, Phase I Trial. *Clin Cancer Res* **23**, 3277-3284, doi:10.1158/1078-0432.CCR-16-2299 (2017).
- 15 Habtezion, A., Nguyen, L. P., Hadeiba, H. & Butcher, E. C. Leukocyte Trafficking to the Small Intestine and Colon. *Gastroenterology* **150**, 340-354, doi:10.1053/j.gastro.2015.10.046 (2016).
- 16 Cabrita, R. *et al.* Tertiary lymphoid structures improve immunotherapy and survival in melanoma. *Nature* **577**, 561-565, doi:10.1038/s41586-019-1914-8 (2020).
- 17 Petitprez, F. *et al.* B cells are associated with survival and immunotherapy response in sarcoma. *Nature* **577**, 556-560, doi:10.1038/s41586-019-1906-8 (2020).
- 18 Helmink, B. A. *et al.* B cells and tertiary lymphoid structures promote immunotherapy response. *Nature* **577**, 549-555, doi:10.1038/s41586-019-1922-8 (2020).

- 19 Sautes-Fridman, C., Petitprez, F., Calderaro, J. & Fridman, W. H. Tertiary lymphoid structures in the era of cancer immunotherapy. *Nat Rev Cancer* **19**, 307-325, doi:10.1038/s41568-019-0144-6 (2019).
- 20 Peck, A. R. *et al.* Validation of tumor protein marker quantification by two independent automated immunofluorescence image analysis platforms. *Mod Pathol* **29**, 1143-1154, doi:10.1038/modpathol.2016.112 (2016).
- 21 Gordy, J. T. *et al.* Treatment with an immature dendritic cell-targeting vaccine supplemented with IFN-alpha and an inhibitor of DNA methylation markedly enhances survival in a murine melanoma model. *Cancer Immunol Immunother* **69**, 569-580, doi:10.1007/s00262-019-02471-0 (2020).
- 22 Eerola, A. K., Soini, Y. & Paakko, P. A high number of tumor-infiltrating lymphocytes are associated with a small tumor size, low tumor stage, and a favorable prognosis in operated small cell lung carcinoma. *Clin Cancer Res* **6**, 1875-1881 (2000).
- 23 Fontenot, J. D., Gavin, M. A. & Rudensky, A. Y. Foxp3 programs the development and function of CD4+CD25+ regulatory T cells. *Nat Immunol* **4**, 330-336, doi:10.1038/ni904 (2003).
- 24 Kim, J. M., Rasmussen, J. P. & Rudensky, A. Y. Regulatory T cells prevent catastrophic autoimmunity throughout the lifespan of mice. *Nat Immunol* **8**, 191-197, doi:10.1038/ni1428 (2007).
- 25 Ali, K. *et al.* Inactivation of PI(3)K p110delta breaks regulatory T-cell-mediated immune tolerance to cancer. *Nature* **510**, 407-411, doi:10.1038/nature13444 (2014).
- 26 Putz, E. M. *et al.* PI3Kdelta is essential for tumor clearance mediated by cytotoxic T lymphocytes. *PLoS One* **7**, e40852, doi:10.1371/journal.pone.0040852 (2012).
- 27 Wert-Carvajal, C. *et al.* Predicting MHC I restricted T cell epitopes in mice with NAP-CNB, a novel online tool. *Sci Rep* **11**, 10780, doi:10.1038/s41598-021-89927-5 (2021).
- 28 O'Connor, C. M., Hoffa, M. T., Taylor, S. E., Avelar, R. A. & Narla, G. Protein phosphatase 2A Aalpha regulates Abeta protein expression and stability. *J Biol Chem* **294**, 5923-5934, doi:10.1074/jbc.RA119.007593 (2019).
- 29 Janssens, V. & Goris, J. Protein phosphatase 2A: a highly regulated family of serine/threonine phosphatases implicated in cell growth and signalling. *Biochem J* **353**, 417-439, doi:10.1042/0264-6021:3530417 (2001).
- 30 Janssens, V., Goris, J. & Van Hoof, C. PP2A: the expected tumor suppressor. *Curr Opin Genet Dev* **15**, 34-41, doi:10.1016/j.gde.2004.12.004 (2005).
- 31 Mumby, M. PP2A: unveiling a reluctant tumor suppressor. *Cell* **130**, 21-24,

- doi:10.1016/j.cell.2007.06.034 (2007).
- 32 Westermarck, J. & Hahn, W. C. Multiple pathways regulated by the tumor suppressor PP2A in transformation. *Trends Mol Med* **14**, 152-160, doi:10.1016/j.molmed.2008.02.001 (2008).
- 33 Freeman, A. K. & Monteiro, A. N. Phosphatases in the cellular response to DNA damage. *Cell Commun Signal* **8**, 27, doi:10.1186/1478-811X-8-27 (2010).
- 34 Chowdhury, D. *et al.* gamma-H2AX dephosphorylation by protein phosphatase 2A facilitates DNA double-strand break repair. *Mol Cell* **20**, 801-809, doi:10.1016/j.molcel.2005.10.003 (2005).
- 35 Lu, J. *et al.* Inhibition of serine/threonine phosphatase PP2A enhances cancer chemotherapy by blocking DNA damage induced defense mechanisms. *Proc Natl Acad Sci U S A* **106**, 11697-11702, doi:10.1073/pnas.0905930106 (2009).
- 36 Cannavo, E., Gerrits, B., Marra, G., Schlapbach, R. & Jiricny, J. Characterization of the interactome of the human MutL homologues MLH1, PMS1, and PMS2. *J Biol Chem* **282**, 2976-2986, doi:10.1074/jbc.M609989200 (2007).
- 37 Speybroeck, N. Classification and regression trees. *Int J Public Health* **57**, 243-246, doi:10.1007/s00038-011-0315-z (2012).
- 38 Samstein, R. M. *et al.* Tumor mutational load predicts survival after immunotherapy across multiple cancer types. *Nat Genet* **51**, 202-206, doi:10.1038/s41588-018-0312-8 (2019).
- 39 Ahmed, D. *et al.* Epigenetic and genetic features of 24 colon cancer cell lines. *Oncogenesis* **2**, e71, doi:10.1038/oncsis.2013.35 (2013).
- 40 Fay, M. P. & Proschan, M. A. Wilcoxon-Mann-Whitney or t-test? On assumptions for hypothesis tests and multiple interpretations of decision rules. *Stat Surv* **4**, 1-39, doi:10.1214/09-SS051 (2010).
- 41 Guinney, J. *et al.* The consensus molecular subtypes of colorectal cancer. *Nat Med* **21**, 1350-1356, doi:10.1038/nm.3967 (2015).
- 42 Tauriello, D. V. F. *et al.* TGFbeta drives immune evasion in genetically reconstituted colon cancer metastasis. *Nature* **554**, 538-543, doi:10.1038/nature25492 (2018).
- 43 Gonzalez-Zuniga, M. *et al.* c-Abl stabilizes HDAC2 levels by tyrosine phosphorylation repressing neuronal gene expression in Alzheimer's disease. *Mol Cell* **56**, 163-173, doi:10.1016/j.molcel.2014.08.013 (2014).
- 44 Shimizu, E., Nakatani, T., He, Z. & Partridge, N. C. Parathyroid hormone regulates histone deacetylase (HDAC) 4 through protein kinase A-mediated phosphorylation and dephosphorylation in osteoblastic cells. *J Biol Chem* **289**, 21340-21350, doi:10.1074/jbc.M114.550699 (2014).
- 45 Kent, L. N. & Leone, G. The broken cycle: E2F dysfunction in cancer. *Nat Rev*

- Cancer* **19**, 326-338, doi:10.1038/s41568-019-0143-7 (2019).
- 46 Mori, K. *et al.* Linkage of E2F1 transcriptional network and cell proliferation with respiratory chain activity in breast cancer cells. *Cancer Sci* **107**, 963-971, doi:10.1111/cas.12953 (2016).
- 47 Yao, G., Lee, T. J., Mori, S., Nevins, J. R. & You, L. A bistable Rb-E2F switch underlies the restriction point. *Nat Cell Biol* **10**, 476-482, doi:10.1038/ncb1711 (2008).
- 48 Kwon, J. S. *et al.* Controlling Depth of Cellular Quiescence by an Rb-E2F Network Switch. *Cell Rep* **20**, 3223-3235, doi:10.1016/j.celrep.2017.09.007 (2017).
- 49 Luchini, C. *et al.* Comprehensive characterisation of pancreatic ductal adenocarcinoma with microsatellite instability: histology, molecular pathology and clinical implications. *Gut* **70**, 148-156, doi:10.1136/gutjnl-2020-320726 (2021).
- 50 Michel, L. L. *et al.* Immune Checkpoint Blockade in Patients with Triple-Negative Breast Cancer. *Target Oncol* **15**, 415-428, doi:10.1007/s11523-020-00730-0 (2020).
- 51 Henriksen, A., Dyhl-Polk, A., Chen, I. & Nielsen, D. Checkpoint inhibitors in pancreatic cancer. *Cancer Treat Rev* **78**, 17-30, doi:10.1016/j.ctrv.2019.06.005 (2019).
- 52 Marcus, L., Lemery, S. J., Keegan, P. & Pazdur, R. FDA Approval Summary: Pembrolizumab for the Treatment of Microsatellite Instability-High Solid Tumors. *Clin Cancer Res* **25**, 3753-3758, doi:10.1158/1078-0432.CCR-18-4070 (2019).
- 53 Le, D. T. *et al.* Mismatch repair deficiency predicts response of solid tumors to PD-1 blockade. *Science* **357**, 409-413, doi:10.1126/science.aan6733 (2017).
- 54 Suzuki, H. *et al.* Intratumoral CD8(+) T/FOXP3 (+) cell ratio is a predictive marker for survival in patients with colorectal cancer. *Cancer Immunol Immunother* **59**, 653-661, doi:10.1007/s00262-009-0781-9 (2010).
- 55 Ling, A., Edin, S., Wikberg, M. L., Oberg, A. & Palmqvist, R. The intratumoural subsite and relation of CD8(+) and FOXP3(+) T lymphocytes in colorectal cancer provide important prognostic clues. *Br J Cancer* **110**, 2551-2559, doi:10.1038/bjc.2014.161 (2014).
- 56 Yoon, H. H. *et al.* Prognostic impact of FoxP3+ regulatory T cells in relation to CD8+ T lymphocyte density in human colon carcinomas. *PLoS One* **7**, e42274, doi:10.1371/journal.pone.0042274 (2012).

Reviewers' Comments:

Reviewer #2:

Remarks to the Author:

The authors have added additional data to address my concerns. The authors now show that PP2A pull-down indeed induces reduced phosphorylation of the proposed targets.

The authors argue that human CRC might have higher PP2A activity which can not be true. But I do think that due to the intratumor heterogeneity in genetic or epigenetic events leading to PP2A inactivation, it is likely that some tumor cells retain a functional PP2A activity and inhibition of which induces MSI status in PP2A functional tumor cells and thus sensitizes immunotherapy. Given that ppp2r2b is silenced in majority of CRC tumors (but in different levels in CRC tumors), applying PP2A inhibitor in CRC tumors with low levels of PPP2R2B methylation and thus more "normal" PP2A activity might be more meaningful with respect to MSI induction and increases response to immunotherapy. A discussion in this context might be helpful to clarify and justify the strategy to inhibit PP2A.

Reviewer #3:

Remarks to the Author:

Thank you for carefully addressing my comments. In my view, the arguments made in the paper are strengthened by the new data. It will be interesting to determine in future studies whether the putative neoepitopes are indeed recognized by relevant T cells (i.e. those which reject the tumour).

Reviewer #4:

Remarks to the Author:

Yen et al. provide a revised manuscript detailing in vitro, in vivo, and correlative bioinformatic analyses to evaluate the role of PP2A activity in the acquisition of mismatch repair deficiency.

As before, the study explores an interesting and certainly relevant concept. Overall, the revised manuscript addresses most of the concerns and questions raised by reviewers, with the following exceptions:

1. In Figure 5e and 5d, the authors have shown that PP2A inactivation with small-molecule inhibitors (LB100, LB102) or shRNA knockdown is associated with loss of MLH1 expression in MSS CRC cell line models. Induction of MSI is seen in the treated samples. However, details about the experimental design are lacking: how long were the cells exposed to LB100 or LB102 (continuous over 2 to 7 days)? How long (days?) after shRNA knockdown did authors evaluate MSI induction? Were the control (CTR) cells analyzed at a single timepoint only? Was the shRNA scramble control used as well? For comparison, it would also be of interest to also include a positive control represented by a cell line with known dMMR/MSI-H.

2. In Figure 2, the authors examine patterns of MSI induction and transcriptomic changes in Lgr5-EGFP-CreERT2; Ppp2r1a flox/flox intestinal organoids treated with DMBA, MNU, or PhIP. Details about the experimental design are lacking: what was the exposure dose to DMBA, MNU, or PhIP? In addition, Figure 2f appears to show that MLH1 expression is decreased with exposure to MNU and PhIP alone – can the authors explain this?

3. In the abstract, the authors state that "the mechanism of MSI status development is not well understood". This revised statement still does not accurately reflect the current understanding of MSI. It should be clarified that MSI is a molecular phenotype caused by mismatch repair deficiency (MMRd). This is commonly explained with a "two-hit" model where MMR genes MLH1, MSH2, MSH6, PMS2 undergo biallelic inactivation. For example in Lynch syndrome there is a pathogenic germline mutation and the neoplastic cells undergo somatic inactivation of the second functional

allele or loss-of-heterozygosity. More often, sporadic MMRd occurs through somatic bi-allelic inactivation of MMR genes through epigenetic silencing, SNV/deletion and LOH. What the authors here are describing is potentially a novel mechanism of acquired MMR deficiency that is induced by PP2A inactivation.

4. Regarding Figure 3g, the authors state in the rebuttal letter that they have “double checked the data and have now provided clear E2F1 data”. Can the authors please clarify what is meant by “double-checked the data”? What was the source of the discrepancy? Was the experiment repeated?

5. Prior to the use of immune checkpoint blockade, MSI-H/dMMR was historically associated with poorer outcomes in advanced (stage IV) colorectal cancer (which is inverted compared to its association with favorable outcomes in early stage I-III disease). In light of this, the statements on page 3 line 54-55 and 64 (“High incidence of somatic mutations can lead to MSI tumours of a less aggressive nature”) are not entirely accurate and should be revised.

6. The authors claim that PPP2R1A, SET, and CIP2A mutations “help to predict responses to ICB” is not strongly supported by the results. Namely, univariate Cox regression analysis of survival differences between PPP2R1A-mutated and PPP2R1A-non-mutated tumor is insufficient and lacks adjustment for other prognostic / predictive factors. MSI status and tumor mutation burden (TMB) are themselves predictive biomarkers, but the authors haven’t demonstrated that PPP2R1A-mutation status is independent of these.

REVIEWER COMMENTS

Reviewer #2 (Remarks to the Author):

The authors have added additional data to address my concerns. The authors now show that PP2A pulldown indeed induces reduced phosphorylation of the proposed targets.

The authors argue that human CRC might have higher PP2A activity which can not be true. But I do think that due to the intratumor heterogeneity in genetic or epigenetic events leading to PP2A inactivation, it is likely that some tumor cells retain a functional PP2A activity and inhibition of which induces MSI status in PP2A functional tumor cells and thus sensitize immunotherapy. Given that PPP2R2B is silenced in majority of CRC tumors (but in different levels in CRC tumors), applying PP2A inhibitor in CRC tumors with low levels of PPP2R2B methylation and thus more "normal" PP2A activity might be more meaningful with respect to MSI induction and increases response to immunotherapy. A discussion in this context might be helpful to clarify and justify the strategy to inhibit PP2A.

Response: Thank you very much for your great comments. According to your suggestion, we have now addressed this in the discussion section, as "Although it has been reported that some subunits of the PP2A holoenzyme, such as PPP2R2B¹, are highly methylated and silenced in some CRC tumours, PP2A inhibition may not induce MSI status in these tumours. Due to intratumoural heterogeneity of genetic or epigenetic events leading to PP2A inactivation, some tumour cells may retain functional PP2A activity. For these PP2A functional tumour cells, PP2A inhibition may induce MSI status, thereby making them sensitive to ICB." in the revised version. Please refer to Page 17 in Line 380-385.

Reviewer #3 (Remarks to the Author):

Thank you for carefully addressing my comments. In my view, the arguments made in the paper are strengthened by the new data. It will be interesting to determine in future studies whether the putative neoepitopes are indeed recognised by relevant T cells (i.e. those which reject the tumour).

Response: Thank you very much for your great comments.

Reviewer #4 (Remarks to the Author): to replace #1

Yen et al. provide a revised manuscript detailing in vitro, in vivo, and correlative bioinformatic analyses to evaluate the role of PP2A activity in the acquisition of mismatch repair deficiency.

As before, the study explores an interesting and certainly relevant concept. Overall, the revised manuscript addresses most of the concerns and questions raised by reviewers, with the following exceptions:

1. In Figure 5e and 5d, the authors have shown that PP2A inactivation with small-molecule inhibitors (LB100, LB102) or shRNA knockdown is associated with loss of MLH1 expression in MSS CRC cell line models. Induction of MSI is seen in the treated samples. However, details about the experimental design are lacking: how long were the cells exposed to LB100 or LB102 (continuous over 2 to 7 days)? How long (days?) after shRNA knockdown did authors evaluate MSI induction? Were the control (CTR) cells analyzed at a single timepoint only? Was the shRNA scramble control used as well? For comparison, it would also be of interest to also include a positive control represented by a cell line with known dMMR/MSI-H.

Response: Thank you very much for your great comments.

1. HT29 and SW620 were cultured in complete medium by RPMI 1640 and DMEM, respectively with 10% FBS and 1% Penicillin-Streptomycin-Amphotericin B Solution (Biological Industries, USA) at 37°C and 5% CO₂. For treatment with LB100 (CC7693, ChemCatch) and LB102 (a gift from the Medicinal chemistry team in New Drug Development Centre of China Medical University), HT29 or SW620 cells were resuspended in serum-free medium and incubated for 4 h to avoid the sticking of drugs by proteases and albumins in serum, followed by incubation in complete medium without (vehicle control) or with 2.5 μM LB100 or 2.5 μM LB102 with medium change every two days, when vehicle, LB100 or LB102 were continuously added. Cells were then harvested, and cell pellets were used for western blotting and MSI status analysis at 2 (D2) and 7 days (D7). Control (vehicle) cells were analyzed at 7 days. We have also modified the M&M in the revised version. Please refer to Page 22 in Line 495-505.

2. Cells were transfected with scrambled shRNAs (control) or shRNAs against PPP2R1A by Lipofectamine 2000 following manufacturer's recommendations (Invitrogen/Life Technologies, USA). After 48 hours transfection, transfected cells were selected by puromycin antibiotics (2 μg/ml) for one week or until complete eradication of non-transfected control cells. Cells were resuspended and reseeded

for removing dead cells. Cells were then harvested and cell pellets were used for western blotting and MSI status analysis (Fig. 5e, S11). We have also modified the M&M in the revised version. Please refer to Page 25 in Line 576-580.

3. Control (vehicle) cells were analyzed at 7 days.

4. We have now added the MSI status data of cells transfected scrambled shRNA in the revised version (Fig. 5e, S11, and S12).

5. Regarding the question “For comparison, it would also be of interest to also include a positive control represented by a cell line with known dMMR/MSI-H.”, we first showed that treatment of mouse cell lines CT26 (MSS) and MC38 (MSI) with LB100 induced MLH1 loss (Fig. S8a) and an increase in MSI status (Fig. S8b). For comparison of MSI by analysis of microsatellite markers in CT26 and MC38 control and LB100-treated cells, LB100 were treated for 2 days. The amplicon bands, predominant amplicon size (peaks) in alleles of cells without (CTR) or with treatment with LB100 are indicated as black and red lines, number(s) and asterisk(s) in all sample traces, respectively. Shiftings comparing to CTR are marked at right at each mononucleotide region panels. The mononucleotide regions used to evaluate microsatellite instability included mBAT-26, mBAT-24, mBAT-37 and L24372.

2. In Figure 2, the authors examine patterns of MSI induction and transcriptomic changes in Lgr5-EGFP-CreERT2; Ppp2r1a flox/flox intestinal organoids treated with DMBA, MNU, or PhIP. Details about the experimental design are lacking: what was the exposure dose to DMBA, MNU, or PhIP? In addition, Figure 2f appears to show

that MLH1 expressed is decreased with exposure to MNU and PhIP alone – can the authors explain this?

Response: Thank you very much for your great comments. *In vitro* culture of intestinal organoids treated with MNU or PhIP as demonstrated by previous study². “*Lgr5-EGFP-CreERT2; Ppp2r1a^{flox/flox}*” intestinal organoids were treated with 5 µg/ml DMBA, 50 µg/ml MNU, or 10 µg/ml PhIP in combination with or without tamoxifen (TAM) for 50 days.” Please refer to Page 38 in Line 862-864.

Regarding the reason that that MLH1 expressed is decreased with exposure to MNU and PhIP alone, our answer is listed below.

It has been reported that cells treated with alkylating agents, such as MNU, underwent DNA mismatch repair (MMR) loss, which in turn led to MSI status³. PhIP has also been reported to cause MSI in normal colon organoid culture⁴.

3. In the abstract, the authors state that “the mechanism of MSI status development is not well understood” . This revised statement still does not accurately reflect the current understanding of MSI. It should be clarified that MSI is a molecular phenotype caused by mismatch repair deficiency (MMRd). This is commonly explained with a “two-hit” model where MMR genes MLH1, MSH2, MSH6, PMS2 undergo biallelic inactivation. For example in Lynch syndrome there is a pathogenic germline mutation and the neoplastic cells undergo somatic inactivation of the second functional allele or loss-of-heterozygosity. More often, sporadic MMRd occurs through somatic bi-allelic inactivation of MMR genes through epigenetic silencing, SNV/deletion and LOH. What the authors here are describing is potentially a novel mechanism of acquired MMR deficiency that is induced by PP2A inactivation.

Response: Thank you very much for the comments. We have now revised the abstract as “Microsatellite-unstable (MSI), a predictive biomarker for immune checkpoint blockade (ICB) response, is caused by mismatch repair deficiency (MMRd) that occurs through genetic or epigenetic silencing of MMR genes. Here, we report a novel mechanism of MMRd and demonstrate that.....” We have also modified the first paragraph in Introduction by adding this information. “... which mainly occur through epigenetic silencing that inactivates the somatic biallelic of the MMR genes⁵.” Please refer to Page 2 in Line 32-35 and Page 3 in Line 55-56.

4. Regarding Figure 3g, the authors state in the rebuttal letter that they have “double checked the data and have now provided clear E2F1 data” . Can the authors please clarify what is meant by “double-checked the data” ? What was the source of the discrepancy? Was the experiment repeated?

Response: Thank you very much for your great comments. We checked the data

again (that is what we mean “double-checked”) and found out that when we extended the exposure time from 4 sec to 20 sec (data were shown below), we found that E2F1 knockdown has a more pronounced effect on the change in E2F1 level, and it has a stronger correlation with changes in downstream gene levels. The left panel was shown in the revision version, while the right panel was shown in the original version.

5. Prior to the use of immune checkpoint blockade, MSI-H/dMMR was historically associated with poorer outcomes in advanced (stage IV) colorectal cancer (which is inverted compared to its association with favorable outcomes in early stage I-III disease). In light of this, the statements on page 3 line 54-55 and 64 (“High incidence of somatic mutations can lead to MSI tumours of a less aggressive nature”) are not entirely accurate and should be revised.

Response: Thank you very much for your great comments. We have now revised the Main Text as “MSI is associated with better stage-adjusted prognosis in early stage I-III colorectal cancer⁶ and response to immune checkpoint blockade (ICB)⁷ than microsatellite-stable (MSS) tumours, leading to the urgent need to investigate the mechanisms causing MSI tumour development.” Please refer to Page 3 in Line 56-59. We have also deleted “High incidence of somatic mutations can lead to MSI tumours of a less aggressive nature”.

6. The authors claim that PPP2R1A, SET, and CIP2A mutations “help to predict responses to ICB” is not strongly supported by the results. Namely, univariate Cox regression analysis of survival differences between PPP2R1A-mutated and PPP2R1A-non-mutated tumor is insufficient and lacks adjustment for other prognostic / predictive factors. MSI status and tumor mutation burden (TMB) are

themselves predictive biomarkers, but the authors haven't demonstrated that PPP2R1A-mutation status is independent of these.

Response: Thank you very much for your great comments. Based on your comment, we have checked the correlation between patient survival and all parameters of the cohort. Among them, total mutation burden (TMB) and PPP2R1A mutation were found to significantly increase the survival rate of patients and reduce the hazard ratio (HR). We then compared the numbers of TMB between PPP2R1A-mutated and PPP2R1A-non-mutated tumors. Besides, we further compared the numbers of TMB between tumors with and without mutation in several driver mutations, such as TP53, PIK3CA and KRAS. We found that TMB was significantly higher in PPP2R1A-mutated than PPP2R1A-non-mutated tumors ($p = 0.00026$). TMB was also significantly higher in TP53 ($p = 2.92 \times 10^{-6}$) and PIK3CA ($p = 0.007$) mutation groups than non-mutation groups (Unshown Table 1). There was a tendency for TMB in KRAS mutated tumors to be higher than in KRAS non-mutated tumors. These data indicate that tumors with high TMB are more likely to have some key driver mutations than tumors with low TMB. When performed univariate Cox regression analysis of survival differences between gene-mutated and gene-non-mutated tumors, we found that only PPP2R1A mutation reduced the HR to 0.4296 ($p = 0.03$), while TP53 and PIK3CA mutations significantly increased the HR to 1.473 and 1.31, respectively (Unshown Table 2). There was a tendency for KRAS mutation to increase HR to 1.31. These data indicate that, except for PPP2R1A, most single-gene mutations did not reduce HR, and cannot be used as "favorable prognostic markers" to help predict the response to ICB (Unshown Figure 1). We further performed adjustment for PPP2R1A mutation and other prognostic / predictive factors. Nevertheless, we also found that the PPP2R1A mutation reduced the HR to 0.6142, although the p-value is not significant (Unshown Table 3).

In addition, we showed in mouse tumor models that loss of PPP2R1A led to increased TMB and MSI status, and in mouse and human tumor cell models, PP2A inactivation also led to loss of MLH1 and induces MSI status (Fig. 5). Therefore, this manuscript can suggest a causal relationship between the PPP2R1A mutation and TMB.

Unshown Table 1. Mean (SEM) and t test for comparison of PPP2R1A mutation in TMB and Age

	mean (SEM)		t-test
	PPP2R1A mutated, n=24	PPP2R1A non-mutated, n=1637	p value
TMB	39.10 (6.48)	11.16 (0.44)	2.6×10^{-5}
Age	61.96 (2.81)	61.40 (0.34)	0.844

	TP53 mutated, n=744	TP 53 non-mutated, n=917	p value
TMB	13.9 (0.79)	9.67 (0.49)	2.92x10 ⁻⁶
Age	61.54(13.7)	51.25 (13.6)	0.662
	PIK3CA mutated, n=200	PIK3CA non-mutated, n=1461	p value
TMB	15.95 (2.23)	9.71 (0.39)	0.007
Age	62.24 (1.4)	61.24 (0.373)	0.509
	KRAS mutated, n=226	KRAS non-mutated, n=1435	p value
TMB	11.05 (1.39)	10 (0.41)	0.495
Age	64.45 (1.221)	61.05(0.376)	0.015

Unshown Table 2. Univariate analysis for overall survival

Variable	HR (95% CI)	P value
PPP2R1A mutation	0.4296 (0.19-0.96)	0.03
TP53 mutation	1.473 (1.26-1.72)	1.5x10 ⁻⁶
PIK3CA mutation	1.4 (1.03-1.895)	0.029
Kras mutation	1.31 (0.98-1.74)	0.067
TMB	0.9822 (0.976-0.988)	4x10 ⁻⁹

HR, hazard ratio; CI, confidence interval.

Unshown Table 3. Multivariate analysis for overall survival

Variable	HR (95% CI)	P value
PPP2R1A mutation	0.6142 (-0.196-1.424)	0.238
TMB	0.9827 (0.97-0.988)	2x10 ⁻⁸

HR, hazard ratio; CI, confidence interval.

References

- 1 Tan, J. *et al.* B55beta-associated PP2A complex controls PDK1-directed myc signaling and modulates rapamycin sensitivity in colorectal cancer. *Cancer Cell* **18**, 459-471, doi:10.1016/j.ccr.2010.10.021 (2010).
- 2 Yen, Y. T. *et al.* PP2A Deficiency Enhances Carcinogenesis of Lgr5(+) Intestinal Stem Cells Both in Organoids and In Vivo. *Cells* **9**, doi:10.3390/cells9010090 (2019).
- 3 Frayling, I. M. Microsatellite instability. *Gut* **45**, 1-4, doi:10.1136/gut.45.1.1 (1999).
- 4 Devall, M. *et al.* Novel insights into the molecular mechanisms underlying risk of colorectal cancer from smoking and red/processed meat carcinogens by modeling exposure in normal colon organoids. *Oncotarget* **12**, 1863-1877, doi:10.18632/oncotarget.28058 (2021).
- 5 Veigl, M. L. *et al.* Biallelic inactivation of hMLH1 by epigenetic gene silencing, a novel mechanism causing human MSI cancers. *Proc Natl Acad Sci U S A* **95**, 8698-8702, doi:10.1073/pnas.95.15.8698 (1998).
- 6 Gryfe, R. *et al.* Tumor microsatellite instability and clinical outcome in young patients with colorectal cancer. *N Engl J Med* **342**, 69-77, doi:10.1056/NEJM200001133420201 (2000).
- 7 Le, D. T. *et al.* Mismatch repair deficiency predicts response of solid tumors to PD-1 blockade. *Science* **357**, 409-413, doi:10.1126/science.aan6733 (2017).

- 8 Tsagris, M. & Pandis, N. Multicollinearity. *Am J Orthod Dentofacial Orthop* **159**, 695-696, doi:10.1016/j.ajodo.2021.02.005 (2021).

Reviewers' Comments:

Reviewer #4:

Remarks to the Author:

Yen et al. kindly provide a revised manuscript in response to reviewers' questions and concerns, which have been adequately addressed.

REVIEWERS' COMMENTS

Reviewer #4 (Remarks to the Author):

Yen et al. kindly provide a revised manuscript in response to reviewers' questions and concerns, which have been adequately addressed.

Response: Thank you very much for your great comments.